# Homeodomain-interacting protein kinase maintains neuronal homeostasis during normal *Caenorhabditis elegans* aging and systemically regulates longevity from serotonergic and GABAergic neurons

**Maria I Lazaro-Pena[1], Adam B Cornwell[1], Carlos A Diaz-Balzac[2], Ritika Das[1,3,4], Zachary C Ward[1], Nicholas Macoretta[3], Juilee Thakar[1,5,6], Andrew V Samuelson[1]\***

[1]Department of Biomedical Genetics, University of Rochester Medical Center, Rochester, United States; [2]Division of Endocrinology, Diabetes and Metabolism, Department of Medicine, University of Rochester Medical Center, Rochester, United States; [3]Department of Biology, University of Rochester, Rochester, United States; [4]Department of Cell Biology, Skirball Institute of Biomolecular Medicine, NYU School of Medicine, New York, United States; [5]Department of Biostatistics and Computational Biology, University of Rochester Medical Center, Rochester, United States; [6]Department of Microbiology and Immunology, University of Rochester Medical Center, Rochester, United States

**\*For correspondence:**
Andrew_Samuelson@URMC.
Rochester.edu

**Competing interest:** The authors declare that no competing interests exist.

**Abstract** Aging and the age-associated decline of the proteome is determined in part through neuronal control of evolutionarily conserved transcriptional effectors, which safeguard homeostasis under fluctuating metabolic and stress conditions by regulating an expansive proteostatic network. We have discovered the *Caenorhabditis elegans* homeodomain-interacting protein kinase (HPK-1) acts as a key transcriptional effector to preserve neuronal integrity, function, and proteostasis during aging. Loss of *hpk-1* results in drastic dysregulation in expression of neuronal genes, including genes associated with neuronal aging. During normal aging *hpk-1* expression increases throughout the nervous system more broadly than any other kinase. Within the aging nervous system, *hpk-1* induction overlaps with key longevity transcription factors, which suggests that *hpk-1* expression mitigates natural age-associated physiological decline. Consistently, pan-neuronal overexpression of *hpk-1* extends longevity, preserves proteostasis both within and outside of the nervous system, and improves stress resistance. Neuronal HPK-1 improves proteostasis through kinase activity. HPK-1 functions cell non-autonomously within serotonergic and γ-aminobutyric acid (GABA)ergic neurons to improve proteostasis in distal tissues by specifically regulating distinct components of the proteostatic network. Increased serotonergic HPK-1 enhances the heat shock response and survival to acute stress. In contrast, GABAergic HPK-1 induces basal autophagy and extends longevity, which requires *mxl-2* (MLX), *hlh-30* (TFEB), and *daf-16* (FOXO). Our work establishes *hpk-1* as a key neuronal transcriptional regulator critical for preservation of neuronal function during aging. Further, these data provide novel insight as to how the nervous system partitions acute and chronic adaptive response pathways to delay aging by maintaining organismal homeostasis.

## Editor's evaluation

This fundamental study substantially advances our understanding of how aging and stress resilience across an organism is determined by identifying a new player in this process and uncovering its mode of action. The evidence is solid as the methods, data and analyses broadly support the claims, with only minor weaknesses. The work will be of broad interest to the field of aging and protein homeostasis.

## Introduction

The gradual decline of function within the proteome (proteostasis) is a characteristic of aging, which precipitates the onset and progression of a growing number of age-associated diseases (*Balch et al., 2008*; *Gidalevitz et al., 2006*; *Gidalevitz et al., 2010*; *Kikis et al., 2010*; *Morimoto, 2011*; *Sala et al., 2017*; *Taylor and Dillin, 2011*): Alzheimer's disease (AD), Parkinson's disease (PD), Huntington's disease, and amyotrophic lateral sclerosis (ALS) are neurodegenerative diseases driven by genetic alterations that typically predispose a mutant protein isoform to aggregate and have toxic gain of function properties on the neuronal proteome. Outside of the nervous system, age-associated proteostatic decline also leads to disease, for example, the onset of diabetes mellitus (*Jaisson and Gillery, 2014*); thus, the decline in proteostasis during normal aging is not limited to the nervous system. A growing number of studies suggest that aging is not simply the result of stochastic accumulation of damage, but is determined through genetics and coordinated mechanisms across tissues and cell types (*Labbadia and Morimoto, 2015*; *Lazaro-Pena et al., 2022*; *López-Otín et al., 2013*). Proteostatic decline corresponds with the age-associated breakdown of a large proteostasis network (PN), which integrates stress-responsive control of protein folding, degradation, and translation in response to myriad cell intrinsic and non-autonomous signals. And yet, while major components of the proteostatic network have been identified, how multi-cellular organisms maintain systemic proteostasis within and across tissues to delay aging is still poorly understood.

The relatively simple metazoan animal *Caenorhabditis elegans* (*C. elegans*) is a premier model system to elucidate how proteostasis is coordinated across cell and tissue types in response to myriad signals. Discoveries of cell non-autonomous signaling first made in *C. elegans* have evolutionarily conserved components, and revealed that the nervous system acts as a key cell non-autonomous regulator of organismal proteostasis and longevity (reviewed in *Miller et al., 2020*). For example, a pair of thermosensory neurons systemically regulate the heat shock response (HSR) in a serotonin-dependent manner (*Prahlad et al., 2008*; *Tatum et al., 2015*). In contrast, a second regulatory component from the GABAergic and cholinergic systems normally limits muscle cell proteostasis (*Garcia et al., 2007*; *Silva et al., 2013*).

The first discovery of a putative link between the transcriptional cofactor homeodomain-interacting protein kinase (HPK-1) and longevity was from our genome-wide RNAi screen that identified 103 genes essential for decreased insulin-like signaling (ILS) to extend longevity (*Samuelson et al., 2007*). In *C. elegans, hpk-1* preserves proteostasis, stress response, and organismal longevity (*Berber et al., 2016*; *Das et al., 2017*; *Samuelson et al., 2007*). Additionally, we have previously shown that HPK-1 extends longevity through distinct genetic pathways defined by HSF-1 and the target of rapamycin complex 1 (TORC1) (*Das et al., 2017*). Activation by either metabolic or genotoxic stressors has been observed from yeast to mammals, suggesting that this family of transcriptional cofactors arose early in evolution to couple metabolic and stress signaling (*Hofmann et al., 2003*; *Kim et al., 2006*; *Lee et al., 2008*; *Shojima et al., 2012*).

In general, mammalian HIPK family members regulate the activity of transcription factors (TFs), chromatin modifiers, signaling molecules, and scaffolding proteins in response to cellular stress (*Calzado et al., 2007*; *de la Vega et al., 2012*; *Song and Lee, 2003a*; *Song and Lee, 2003b*). For example, nutrient stress, such as glucose deprivation, can activate Hipk1 and Hipk2 (*Ecsedy et al., 2003*; *Garufi et al., 2013*; *Song and Lee, 2003b*). Conversely, hyperglycemia triggers HIPK2 degradation via the proteasome (*Baldari et al., 2017*).

We sought to identify the tissue and cell types in which the HPK-1 transcriptional circuits act to extend longevity, increase resistance to acute proteotoxic stress, and which PN components are regulated across tissues. We have discovered that HPK-1 functions as a key regulator of the proteostatic response, originating in the nervous system of *C. elegans*. We find that *hpk-1* is the most broadly upregulated kinase during normal *C. elegans* aging, predominantly within the nervous system, and

**eLife digest** Proteins are essential for nearly every cellular process to sustain a healthy organism. A complex network of pathways and signalling molecules regulates the proteins so that they work correctly in a process known as proteostasis.

As the body ages, this network can become damaged, which leads to the production of faulty proteins. Many proteins end up being misfolded – in other words, they are misshapen on the molecular level, which can be toxic for the cell. A build-up of such misfolded proteins is implicated in several neurological conditions, including Alzheimer's, Parkinson's and Huntington's disease.

Cells have various ways to detect and respond to internal stressors, such as tissue or organ damage. For example, specific proteins in the nervous system can raise a 'central' alert when damage is detected, which then primes and coordinates the body's systems to respond in the peripheral cells and tissues. But exactly how this happens is still unclear.

To find out more about the central coordination of stress responses, Lazaro-Pena et al. studied one such sensor protein, called HPK-1, in the roundworm *C. elegans*. They first overexpressed the protein in various tissues. This revealed that only when HPK-1 was overactive in nerve tissue, it protected proteins and prolonged the lifespan of the worms. An increased amount of HPK-1 improved the health span of the worms and older worms also moved better. However, genetically manipulated worms lacking HPK-1 in their nerve cells showed a faster decline in nervous system health as they aged, which could be reversed once HPK-1 was activated again.

Lazaro-Pena et al. then measured the amount of HPK-1 in worms at different stages of their life. This showed that as the worms aged, the amount of HPK-1 increased in the nerve cells. The nerve cells in which HPK-1 levels increased overlapped with an increased expression of proteins associated with longevity. Moreover, when HPK-1 was overexpressed, it stimulated the release of other cell signals, which then triggered protective responses to prevent the misfolding and aggregation of proteins and to help degrade damaged proteins.

This study shows for the first time that HPK-1 appears to play a protective role during normal ageing and that it may act as a key switch to stimulate other protective mechanisms. These findings may give rise to new insights into how the nervous system can coordinate many different stress responses, and ultimately delay ageing throughout the whole body.

acts to preserve neuronal function and proteostasis within the nervous system. Further, HPK-1 kinase activity within the nervous system initiates cell non-autonomous signals that trigger protective peripheral responses to maintain organismal homeostasis. Responses can qualitatively differ depending on the neuronal type from which they arise: serotonergic HPK-1 activity increases thermotolerance and protects the proteome from acute stress by enhancing the HSR, without increasing lifespan or altering basal autophagy. In contrast, GABAergic HPK-1 activity induces autophagy and extends longevity, without altering thermotolerance. HPK-1 overexpression is sufficient to induce autophagy gene expression, and autophagosome formation requires *hlh-30* (TFEB), *mxl-2* (MLX), and *daf-16* (FOXO) TFs. We posit that serotonergic HPK-1 signaling responds to acute proteotoxic stress, whereas GABAergic HPK-1 responds to chronic metabolic changes mediated through TORC1 (*Das et al., 2017*). Yet, each of these distinct adaptive responses is sufficient to activate the PN and improve proteostasis outside of the nervous system. Our findings reveal HPK-1 as a key neuronal regulator in coordinating adaptive metabolic and stress response pathways across tissues to delay aging throughout the animal.

## Results

### HPK-1 acts from neuronal tissue to promote longevity and healthspan

To precisely determine where HPK-1 functions in the regulation of longevity, we overexpressed *hpk-1* within neuronal (*rab-3p*), muscle (*myo-3p*), intestine (*ges-1p*), or hypodermal (*dpy-7p*) tissues to identify where activity was sufficient to increase lifespan. Increased neuronal expression of *hpk-1* was sufficient to increase lifespan by 17%, which is comparable to constitutive overexpression of *hpk-1* throughout the soma (*Figure 1A*; *Das et al., 2017*). In contrast, transgenic animals overexpressing *hpk-1* in either muscular, intestinal, or hypodermal cells had no significant change in lifespan

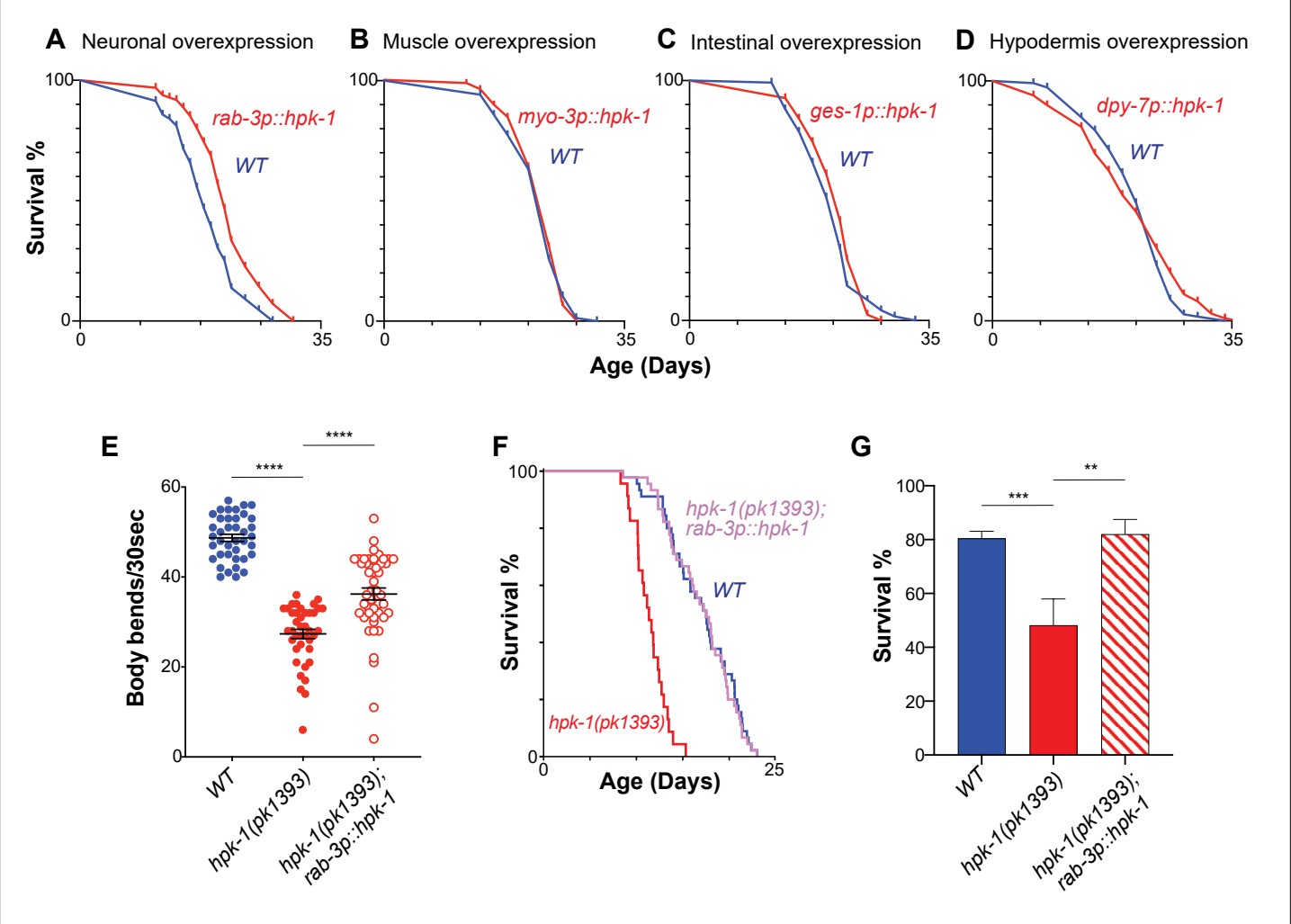

**Figure 1.** Neuronal homeodomain-interacting protein kinase (HPK-1) extends longevity and promotes healthspan. Lifespan curves of animals overexpressing *hpk-1* (red line) in the nervous system (**A**), muscle (**B**), intestine (**C**), and hypodermis (**D**) compared with control non-transgenic siblings (blue line). (n≥175, log-rank test, p<0.0001 for **A**, n≥78 for **B**, n≥82 for **C**, and n≥105 for **D**). Lifespan graphs are representative of two biological replicates. (**E**) Frequency of body bends of *wild-type*, *rab-3p::hpk-1* and *hpk-1(pk1393); rab-3p::hpk-1* day 2 adult animals (n≥48). (**F**) Lifespan curves of wild-type (blue), *hpk-1(pk1393)* (red), and *hpk-1(pk1393); rab-3p::hpk-1* animals (pink). (**G**) Survival of day 1 adult animals subjected to heat shock treatment. Graph represents one of the two individual trials (n≥66). t-Test analysis with **p<0.01, ***p<0.001, and ***p<0.0001. On panels (E and G), bars represent ± SEM. See *Supplementary file 1*, *Supplementary file 2* and *Supplementary file 3* for details and additional trials.

(*Figure 1B–D*). These results were somewhat unexpected, as we previously found that loss of *hpk-1* solely within either neuronal, intestinal, or hypodermal cells is sufficient to shorten lifespan (*Das et al., 2017*). *hpk-1* is broadly expressed during embryogenesis and larval development, but HPK-1 protein is only expressed within the nervous system in adult animals (*Berber et al., 2013*; *Berber et al., 2016*; *Das et al., 2017*; *Raich et al., 2003*). Furthermore, *hpk-1* and orthologs are known to have roles in development, differentiation, and cell fate specification (*Berber et al., 2013*; *Blaquiere and Verheyen, 2017*; *Calzado et al., 2007*; *Hattangadi et al., 2010*; *Rinaldo et al., 2007*; *Rinaldo et al., 2008*; *Steinmetz et al., 2018*). Therefore, we conclude that HPK-1 functions primarily within the nervous system in adult animals to regulate aging.

We performed a locomotion assay to test whether neuronal *hpk-1* could rescue an age-associated decline in movement, a readout of *C. elegans* healthspan (*Berber et al., 2016*; *Das et al., 2017*; *Herndon et al., 2002*; *Huang et al., 2004*; *Samuelson et al., 2007*). As expected, *hpk-1(pk1393)* null mutants displayed a reduced frequency of body bends at day 2 of adulthood (*Figure 1E*). Increased expression of *hpk-1* in neurons was sufficient to partially mitigate the locomotion defect of *hpk-1* null mutants in response to active stimuli. Next, we tested whether restoring *hpk-1* exclusively within

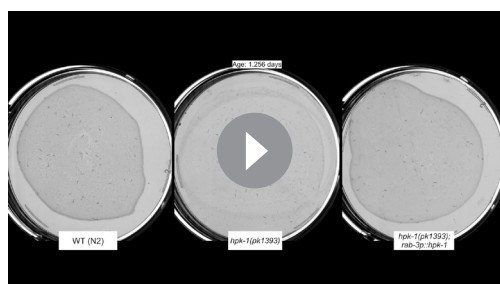

**Video 1.** Homeodomain-interacting protein kinase (HPK-1) functions within the nervous system to regulate healthspan. Video of wild-type, *hpk-1(pk1393)* null animals, and *hpk-1* null animals with neuronal expression of *hpk-1* (*rab-3p::HPK-1*). Videos were generated from consecutive images taken approximately every 15 min over the course of the animals lifespan on 6 cm plates using the Lifespan Machine method (*Stroustrup et al., 2013*).
https://elifesciences.org/articles/85792/figures#video1

the nervous system would prevent the premature age-associated decline in movement of *hpk-1* null mutant animals. Indeed, restoring *hpk-1* neuronal expression rescued the premature age-associated decline in movement of *hpk-1* null mutant animals (*Video 1*). Thus, HPK-1 activity within the nervous system is sufficient to improve healthspan. Lastly, neuronal *hpk-1* rescue was sufficient to restore a wild-type lifespan in *hpk-1* null mutant animals (*Figure 1F*). It should be noted that this same transgene increases wild-type lifespan (*Figure 1A*), which suggests that HPK-1 retains essential functions for adult healthspan and lifespan outside of the nervous system; we posit these functions are related to development as HPK-1 is no longer expressed outside of the nervous system in adult animals (*Das et al., 2017*).

We next asked whether neuronal *hpk-1* expression would be sufficient to rescue the ability of animals to survive acute stress. We and others previously found that in the absence of *hpk-1*, animals are vulnerable to thermal stress (*Berber et al., 2016*; *Das et al., 2017*). Restoring *hpk-1* expression solely within the nervous system was sufficient to rescue the reduced thermotolerance of *hpk-1* null mutant animals (*Figure 1G*). Collectively, these findings are consistent with the notion that *hpk-1* primarily acts within the nervous system to regulate aging and stress resistance.

## *hpk-1* expression increases broadly during normal aging

We sought to determine whether *hpk-1* expression changes during normal aging. *hpk-1* mRNA expression increases in wild-type animals during normal aging between day 2 and 10 of adulthood (*Figure 2A*). Adult *C. elegans* are composed of 959 somatic cells, of which 302 are neuronal; all somatic cells are post-mitotic and arise from an invariant lineage. A recent comprehensive single-cell analysis of age-associated changes in gene expression throughout *C. elegans* identified 211 unique 'cell clusters' within the aging soma: groups of cells within a lineage or tissue defined by gene expression profiles (*Roux et al., 2022*). The '*C. elegans* Aging Atlas' analysis identified 124 neuronal cell clusters and obtained age-associated changes in gene expression for 122, with high concordance of overall gene expression between neuron clusters to a previous project, which mapped the complete gene expression profiles of the *C. elegans* nervous system in late larval animals (The CeNGEN project, *Taylor et al., 2021*). The *C. elegans* Aging Atlas provides the means to investigate age-associated changes in gene expression at an unprecedented level of resolution.

Using the *C. elegans* Aging Atlas dataset, in which differentially expressed genes were identified within each cell cluster in old versus young animals, we find that *hpk-1* expression increases during aging within 48 cell clusters: broadly within the nervous system and in 11 non-neuronal cell clusters (*Figure 2—figure supplement 1*, see Materials and methods). With aging, neuronal expression of *hpk-1* increased within interneurons, motor neurons, sensory neurons, and all neuronal subtypes (cholinergic, dopaminergic, GABAergic, serotonergic, and glutamatergic). We found no cell clusters in which *hpk-1* is downregulated during normal aging. It is well known that many longevity-associated TFs increase lifespan when overexpressed and shorten lifespan when lost (reviewed in *Denzel et al., 2019*), which infers these TFs might be upregulated, or become chronically activated, during normal aging. A major finding of the *C. elegans* Aging Atlas study was that many longevity-associated TFs are upregulated during normal aging in more than 200 cell clusters (*Roux et al., 2022*). Increased expression of longevity-associated TFs is thought to reflect an age-associated adaptive response to dysregulation of homeostatic mechanisms and accumulating damage (*Roux et al., 2022*). We posit *hpk-1* neuronal expression increases during normal aging in wild-type animals to mitigate accumulating damage and cellular dysfunction.

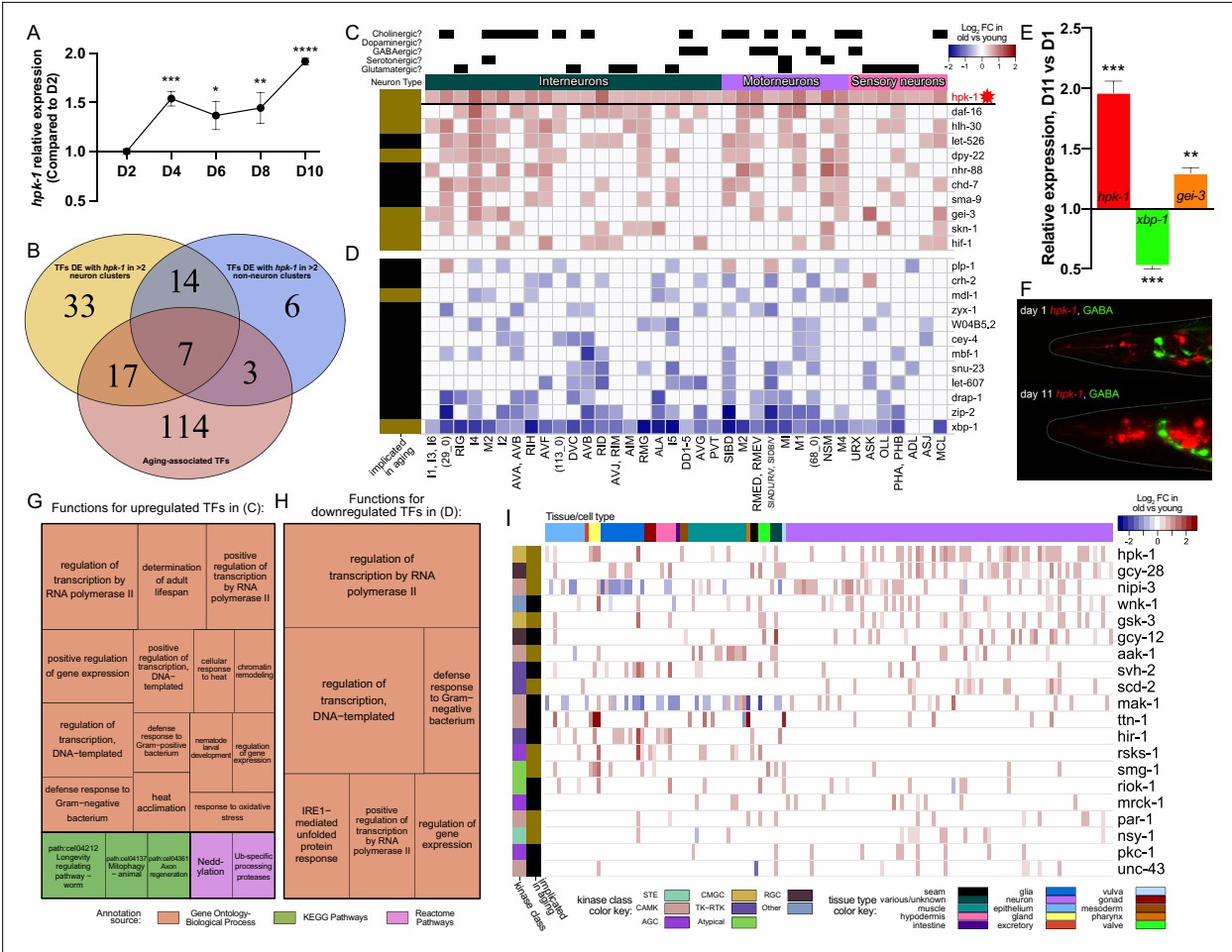

**Figure 2.** During normal aging *hpk-1* is the most broadly upregulated kinase and overlaps with key longevity-associated transcription factors (TFs) within the nervous system. (**A**) Relative expression of *hpk-1* mRNA during normal aging, as measured by RT-qPCR. (**B**) Venn diagram depicting TFs with significant age-associated differential expression alongside *hpk-1* within 3 or more of the 36 neuronal and/or 11 non-neuronal cell clusters (yellow and blue, respectively), and the intersection with TFs that have previously been implicated in *C. elegans* longevity (red). (**C, D**) Heat map of neuronal cell clusters in which *hpk-1* expression increases during normal aging along with TFs that have the most broadly co-occurring differential expression with age (≥9 cell-clusters). Positive (red) and negative (blue) fold-change with aging of a given TF are shown in (**C**) and (**D**), respectively, grouped by average fold-change across the indicated clusters. Neuronal types, subtypes, cell cluster, or individual neuron pair, as well as whether a TF has previously been implicated in aging is indicated. Primary data for this analysis was generated in the *C. elegans* Aging Atlas (*Roux et al., 2022*) and significant differences in expression with aging were determined by filtering differential expression results from a comparison of old vs young animals, based on adjusted p-values <0.05 and a log$_2$ fold-change magnitude of at least 0.5 (between days 8,11,15 and days 1,3,5). (**E**) Relative expression of *hpk-1, xbp-1, and gei-3* mRNA of day 11 adult animals, compared to day 1 as measured by RT-qPCR. (**F**) Representative images of *hpk-1* expression within the nervous system of day 1 and 11 adult animals (*hpk-1p::mCherry; unc-47p::GFP*). See *Figure 7F* and *Figure 7—figure supplement 1A–C* for representative images of whole animals (n≥5). (**G, H**) Functions and pathways associated with two or more of the TFs listed in (**C**) and (**D**), respectively. (**I**) *hpk-1* is differentially expressed with age in more cell clusters than any other kinase, and mostly in neurons. Heat map shows log$_2$ fold-change in old vs young animals for kinases with differential expression in ≥10 cell clusters. Primary data for this analysis was generated in the *C. elegans* Aging Atlas (*Roux et al., 2022*) and filtered for significant age-associated expression changes as described in (**C, D**). On panels (**A** and **E**), bars represent ± SEM of 3 technical replicates with more than 500 animals per condition. t-Test analysis with *p<0.05, **p<0.01, ***p<0.001, and ****p<0.0001. See *Supplementary file 4* for additional RT-qPCR details, and *Supplementary files 5 and 6* for dataset.

The online version of this article includes the following figure supplement(s) for figure 2:

**Figure supplement 1.** During normal aging *hpk-1* is differentially expressed along with transcription factors (TFs) in cell clusters corresponding to tissues outside of the nervous system.

**Figure supplement 2.** *Hpk-1* and all transcription factors (TFs) differentially expressed in overlapping cell clusters within the *C. elegans* nervous system during aging.

**Figure supplement 3.** *hpk-1* is broadly co-expressed with longevity-associated transcription factors (TFs) in late larval stage animals.

**Figure supplement 4.** Only a limited subset of kinases change expression during normal aging.

## During normal aging, *hpk-1* and longevity-associated TFs are upregulated in shared subsets of neurons

We next assessed whether any TFs showed patterns of spatial upregulation similar to the age-associated increase in expression of *hpk-1*, and then segregated those cell types into neuronal and non-neuronal subgroups. Seventy-one TFs were differentially upregulated with *hpk-1* during aging within 3 or more neuron clusters, 30 within non-neuronal clusters, with an overlap of 21 TFs between neurons and non-neurons (*Figure 2—figure supplement 2* and *Supplementary file 5*). Overall, 26 of these TFs have previously been implicated in longevity. There are 10 TFs broadly upregulated with aging in the nervous system that overlap with *hpk-1* (significant differential expression in 9 or more neuronal clusters), 6 of which have previously been linked to longevity: *daf-16* (FOXO, overlaps in 19/37 neuronal clusters), *hlh-30* (TFEB, 18/37), *dpy-22* (mediator complex subunit 12L, 15/37), *gei-3* (10/37), *skn-1* (NRF2, 10/37), and *hif-1* (hypoxia inducible factor 1, 10/37) (*Figure 2C*; *An and Blackwell, 2003*; *Chen et al., 2009*; *Fawcett et al., 2015*; *Kenyon et al., 1993*; *Lapierre et al., 2013*; *Lin et al., 1997*; *Ogg et al., 1997*; *Roux et al., 2022*; *Suriyalaksh et al., 2022*; *Tacutu et al., 2012*; *Zhang et al., 2009*). Conversely, 12 TFs exhibited an inverse correspondence, and were largely downregulated with aging in the same neurons where *hpk-1* is normally upregulated (significant differential expression in nine or more neuronal clusters) (*Figure 2D*); only two of which have previously been implicated in longevity: *xbp-1* (X-box binding protein 1) a key regulator of the endoplasmic reticulum unfolded protein response (ER-UPR), and *mdl-1* (MAX dimerization protein) (*Johnson et al., 2014*; *Nakamura et al., 2016*; *Riesen et al., 2014*). Of note, *mdl-1* is a Myc-family member with opposing roles in longevity from *mxl-2* (Mlx) (*Johnson et al., 2014*), and *mxl-2* is required for *hpk-1* to extend longevity (*Das et al., 2017*).

We tested whether we could verify changes in age-associated gene expression for several candidate longevity-associated TFs identified in the Aging Atlas study. Age-synchronized wild-type animals at either day 1 or 11 of adulthood were harvested and mRNA levels of *hpk-1, xbp-1*, and *gei-3* were assessed. As expected, both *hpk-1* and *gei-1* were significantly induced in older animals, and conversely *xbp-1* expression was decreased (*Figure 2E*). We conclude that we can recapitulate results obtained from the Aging Atlas study and confirm that longevity-associated TFs are significantly differentially expressed during normal aging. Next we determined whether *hpk-1* expression is significantly induced within the *C. elegans* nervous system during normal aging; we assessed age-associated changes in neuronal *hpk-1* mRNA levels in vivo (*hpk-1p::mCherry*). As expected, we find that *hpk-1* is significantly upregulated within isolated *C. elegans* neurons during normal aging (*Figure 2F*).

In non-neuronal clusters we found a much smaller subset of TFs with corresponding positive age-associated fold-changes. Only six of these TFs have been implicated in longevity: *fkh-9* (FOXG), *sea-2*, *zfh-2* (ZFHX3) (*Walter et al., 2011*), *xbp-1*, *mdl-1*, and *hlh-6* (achaete-scute family bHLH TF 3) (*Figure 2—figure supplement 1*).

Within neuronal cell clusters, *hpk-1* and the aforementioned longevity-associated TFs are broadly expressed throughout the nervous system by late larval development, with the exception of the nematode-specific gene *gei-3*, for which overlap is restricted to a subset of neurons (*Figure 2—figure supplement 3*). TFs with age-associated increases in expression in clusters overlapping with *hpk-1* are associated with positive regulation of adult lifespan, heat and oxidative stress response, bacterial innate immunity, chromatin remodeling, development, mitophagy, axon regeneration, neddylation, and ubiquitination (*Figure 2C*). TFs with decreased expression in aging in *hpk-1*-upregulated neuronal cell clusters have been implicated in the ER unfolded protein response and the innate immune response (*Figure 2D*). Interestingly, when assessed at the level of individual time points rather than 'old' and 'young' groups, we find that the age-associated upregulation of *hpk-1* and expression changes of longevity-associated TFs does not necessarily occur in a tightly linked manner suggestive of a single upstream regulatory process (see Appendix 1). Rather, we posit that *hpk-1* is induced within the nervous system along with key longevity TFs during normal aging in response to accumulating stressors and collapsing homeostatic mechanisms to mitigate the physiological effects of aging.

## *hpk-1* is the most broadly upregulated kinase during normal aging

We were surprised to find that *hpk-1* mRNA expression increased during aging, as kinase regulation at the level of gene expression seemed atypical. Therefore, using the *C. elegans* Aging Atlas dataset we assessed whether any of the 438 *C. elegans* kinases (*Zaru et al., 2017*) are differentially expressed

during aging: 115 kinases are upregulated and conversely 120 are downregulated in at least one cell cluster, while 254 kinases do not change expression (*Figure 2—figure supplement 4*, *Supplementary file 6*). *hpk-1* is upregulated in more cell clusters than any other kinase, while *hpk-1* is upregulated in 48, the next most broadly upregulated kinase is *gcy-28* (natriuretic peptide receptor 1, 38 cell clusters) (*Li and van der Kooy, 2018*; *Shinkai et al., 2011*; *Tsunozaki et al., 2008*). Only 14 kinases are downregulated in 10 or more cell clusters. Conversely, 20 kinases are upregulated in 10 or more cell clusters (*Figure 2—figure supplement 4*). In addition to *hpk-1* and *gcy-28*, five kinases previously linked to longevity are upregulated: *nipi-3* (a Tribbles pseudokinase that regulates SKN-1; *Wu et al., 2021*), *aak-1* (AMP-activated kinase catalytic subunit alpha-2) (*Lee et al., 2009*), *scd-2* (ALK receptor tyrosine kinase) (*Shen et al., 2007*), *rsks-1* (ribosomal protein S6 kinase) (*Hansen et al., 2007*; *Pan et al., 2007*), and *smg-1* (SMG1) (*Masse et al., 2008*).

We previously demonstrated that HPK-1 is an essential component of TORC1-mediated longevity and autophagy (*Das et al., 2017*). Interestingly, we find that only 2 of 11 TOR-associated genes, *rsks-1* and *aak-1*, are upregulated during aging in three or more cell clusters (*Blackwell et al., 2019*; *Hindupur et al., 2015*; *Pan et al., 2007*). We find *rsks-1* is exclusively upregulated outside of the nervous system and has poor overlap with *hpk-1* (2/13). *aak-1* is primarily but not exclusively upregulated outside of the nervous system during aging, with some overlap with *hpk-1*. In future studies it will be interesting to dissect whether HPK-1 intersects with these components of TORC1 signaling within and/or outside of the nervous system.

## HPK-1 preserves integrity of the *C. elegans* nervous system

Based on the neuronal HPK-1 capacity to promote longevity, we sought to determine whether HPK-1 preserves the anatomy and physiology of the nervous system during aging. To assess age-associated changes in neuroanatomy of ventral and dorsal cord motor neurons (VD and DD), we used the *unc-25p::GFP* cell-specific reporter (*Figure 3A*; *Cinar et al., 2005*; *Kraemer et al., 2003*). Aged populations of *hpk-1* mutant animals displayed more than a threefold difference in VD axonal breaks and more than a twofold difference in DD axonal breaks compared to wild-type animals (*Figure 3A–B and D*). These age-associated breaks could be rescued by neuronal expression of *hpk-1* (*Figure 3C*). Importantly, young wild-type and *hpk-1* mutant animals did not show neuronal breaks (*Figure 3B*). Thus, loss of *hpk-1* does not disrupt axon formation during development, rather HPK-1 preserves axonal integrity in aging motor neurons.

In *C. elegans*, locomotion is controlled by acetylcholine-releasing excitatory motor neurons and GABA-releasing inhibitory motor neurons (*Jorgensen and Nonet, 1995*; *Richmond and Jorgensen, 1999*). Thus, locomotion is a useful readout to assess neurotransmission. To examine the role of *hpk-1* in synaptic transmission, we performed a paralysis assay after acetylcholinesterase inhibition with aldicarb, where mutations that impair cholinergic signaling cause a reduction in rate of paralysis (*Mahoney et al., 2006*; *Oh and Kim, 2017*). Therefore, mutants with lower synaptic transmission undergo reduced paralysis (i.e., animals are resistant), while mutants with higher synaptic transmission exhibit greater sensitivity. Interestingly, *hpk-1* mutants showed increased aldicarb resistance by L4/YA (*Figure 3E*), suggesting that loss of *hpk-1* reduces cholinergic synaptic transmission and that HPK-1 normally preserves neuronal function.

*C. elegans* serotonin (5-HT) signaling in conjunction with dopamine (DA) modulates locomotion; animals treated with exogenous 5-HT exhibit paralysis over time (*Hardaker et al., 2001*; *Omura et al., 2012*). In a paralysis assay with exogenous 5-HT, we observed that late L4/YA *hpk-1* mutant animals paralyzed sooner: at 2 min of 5-HT treatment, 63% of mutants were immobilized compared to 25% of wild-type animals immobilized (*Figure 3F*). Based on the aforementioned findings, we conclude that HPK-1 prevents an age-associated decline in axonal and synaptic integrity.

## Loss of *hpk-1* results in neuronal dysregulation of neuronal gene expression

We sought to gain deeper insight into how loss of *hpk-1* negatively impacted the *C. elegans* nervous system. To identify early transcriptional dysregulation events that occur due to the absence of *hpk-1*, we conducted bulk RNA-sequencing (RNA-Seq) of day 2 adult wild-type and *hpk-1(pk1393)* null mutant animals, an age at which axonal breaks have yet to occur (*Figure 3B and D*), but after neuronal function has begun to decline (*Figure 3E and F*). Differential expression analysis identified 2201 total

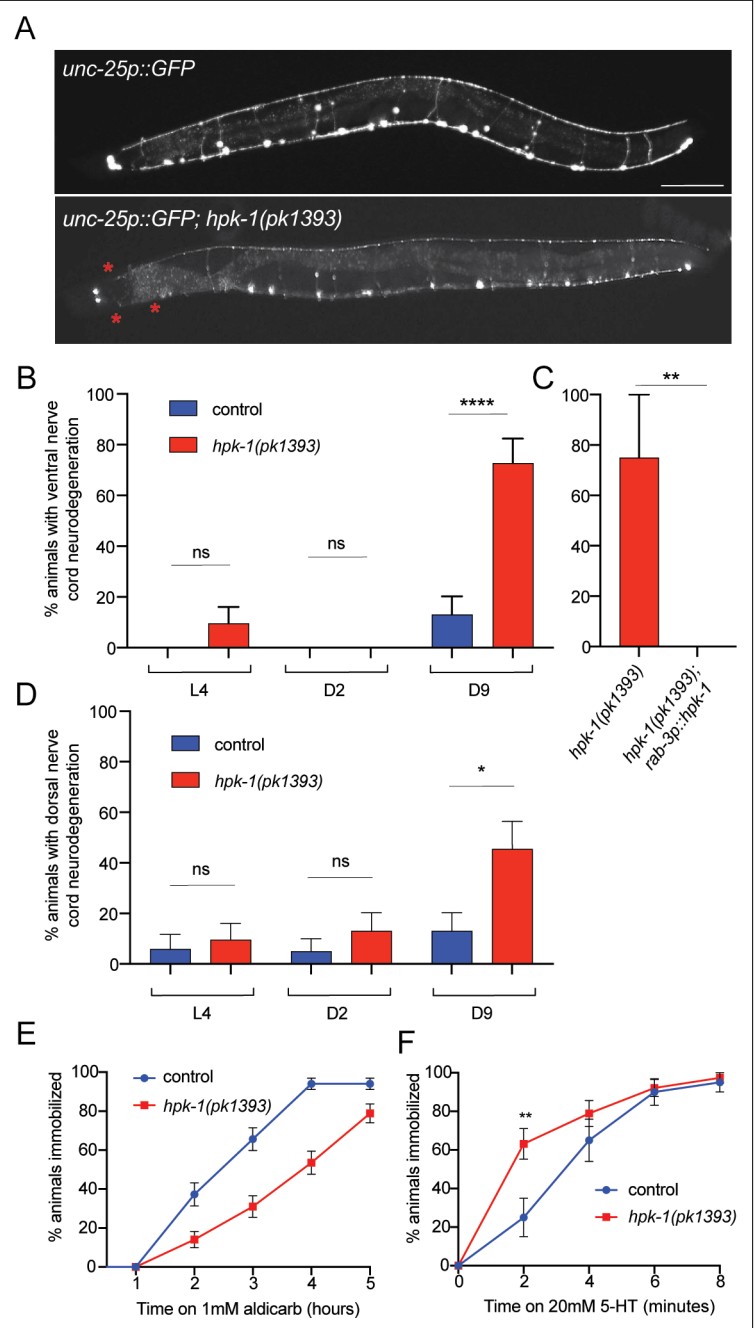

**Figure 3.** Neuronal homeodomain-interacting protein kinase (HPK-1) maintains neuronal integrity during aging. (**A**) Fluorescent micrographs of control and *hpk-1(pk1393)* 9-day-old animals expressing GFP in γ-aminobutyric acid (GABA)ergic DD and VD motor neurons. Asterisks (*) indicate axonal breaks in the motor neurons. Scale bar, 100 μm. (**B**) Percentage of wild-type and *hpk-1* mutant animals with neurodegeneration in VD motor neurons (n≥23). (**C**) Percentage of *hpk-1* rescue animals with neurodegeneration in VD motor neurons (n≥10). (**D**) Percentage of wild-type and *hpk-1* mutant animals with neurodegeneration in DD motor neurons (n≥20). Percentage of immobilized animals after exposure to aldicarb (n≥67) (**D**) and to 20 mM 5-HT (n≥38) (**E**). t-Test analysis with **p<0.01 and ***p<0.001. Bars represent ± SEM. See *Supplementary file 7*, *Supplementary file 8* and *Supplementary file 9* for details and additional trials.

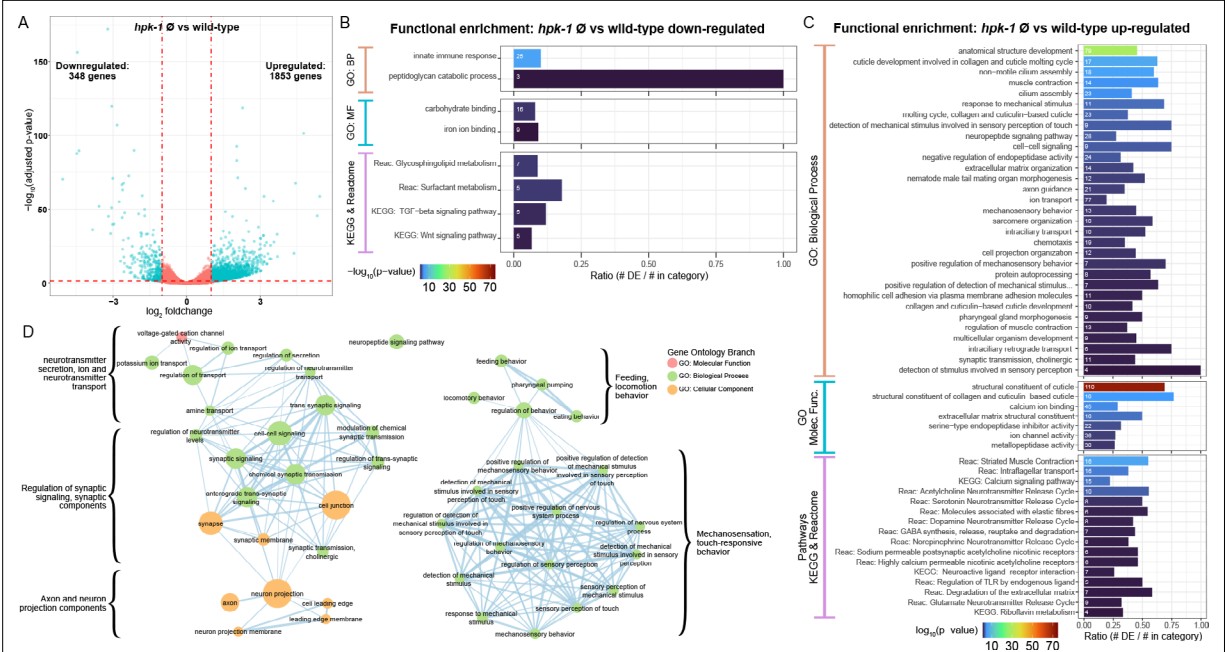

**Figure 4.** Loss of *hpk-1* results in broad dysregulation of neuronal gene expression. (**A**) Differential expression analysis for RNA-sequencing (RNA-Seq) of day 2 adult *hpk-1* null mutant animals (n=3) compared to N2 wild-type animals (n=3) identified 2201 genes with significantly altered expression with loss of *hpk-1* (adjusted p-value <0.05 and |log₂ fold-change|≥1), 84% of which were upregulated in the mutants. The volcano plot illustrates the criteria applied for selecting genes with significant and substantial expression changes, shown as blue dots: vertical dashed lines are the fold-change threshold and the horizontal dashed line is the p-value threshold. (**B, C**) Enrichment for functions and pathways associated with the 348 downregulated genes (**B**) or the 1853 upregulated genes (**C**) in *hpk-1* animals yields innate immune response as the most significantly over-represented term for downregulated genes, but reveals a large number of significant associations to pathways and processed for the upregulated genes – many of which are associated with neurons and neuronal signaling. Each bar represents a functional term or pathway from the Gene Ontology (GO), KEGG, or Reactome databases, as indicated. Bar colors show the –log₁₀ transformed enrichment p-value. The number in each bar is the size of the overlap between the set of differentially expressed genes and the genes in the term or pathway. The x-axis indicates the fraction of the total genes in the term or pathway that were differentially expressed. All results shown have significant enrichment after adjustment for multiple testing. (**D**) Functional enrichment for 283 genes upregulated with *hpk-1* loss that are specific to, or enriched in, *C. elegans* neurons shows broad dysregulation of functions associated with neurons including neurotransmitter transport and release, cell-cell junctions, axons, neuropeptides, and sensory processes, among others; 880 genes uniquely expressed or significantly enriched in neurons were derived from bulk tissue-specific RNA-Seq profiling results in *Kaletsky et al., 2018*, and 32% of these are significantly upregulated in *hpk-1* null animals. See *Supplementary file 10* and *Supplementary file 11* for dataset and additional analysis.

genes with significant expression changes with loss of *hpk-1*, 1853 upregulated and 348 downregulated (adjusted p-value <0.05 and log₂ fold-change magnitude ≥1) (*Figure 4A* and *Supplementary file 10*). Functional enrichment with over-representation analysis was performed for the up- and downregulated genes separately (GOSeq, adjusted p-value <0.01). Among downregulated genes, we find 25 genes implicated in the innate immune response, as well as TGF-beta and WNT signaling (five ubiquitin ligase complex component genes), glycosphingolipid metabolism, and surfactant metabolism (*Figure 4B*). By contrast, more than 50 Gene Ontology (GO) terms and pathways were enriched for the upregulated genes. The most significantly enriched single term reveals a substantial number of cuticle and collagen-associated genes as dysregulated (*Figure 4C*). Upregulation of some collagens and ECM-remodeling processes have been associated with longevity in *C. elegans* (*Ewald et al., 2015*; *Rahimi et al., 2022*). Alternatively, aberrant upregulation could lead to stiffening of the cuticle, a hallmark of aged *C. elegans* and some progeric mutants (*Ewald et al., 2015*; *Rahimi et al., 2022*).

A striking number of other functions significantly enriched in the *hpk-1* upregulated genes from across GO, KEGG, and Reactome are related to neuronal function and signaling, including neuropeptide and neurotransmitter pathways, mechanosensation, chemotaxis, and axon guidance, among others (*Figure 4C*). To explore further, we looked at the *hpk-1* null upregulated genes in the context of a curated compendium of genes with functions that are important for the nervous system (*Hobert, 2013*), as well as genes identified as uniquely expressed in neurons or significantly enriched in neurons from tissue-specific bulk RNA-Seq (*Kaletsky et al., 2018*).

Among 1496 genes known, or expected through homology, to function in the nervous system, 264 are upregulated with loss of *hpk-1*, representing a much larger overlap than expected by chance (hypergeometric test p-value <0.0001) (*Supplementary file 11*). Genes with higher fold-change of expression in *hpk-1* mutants were associated with ciliated sensory neurons (motor proteins, sensory cilia transport), calcium binding, and neuropeptides. We find that 55 of these neuron-associated genes are normally upregulated with age in wild-type animals in at least one cell cluster in the *C. elegans* Aging Atlas, 87 are downregulated with age, with the remaining having mixed or no significant age-associated changes. This suggests that while loss of *hpk-1* results in expression of some genes associated with aging, the bulk of differentially expressed genes at day 2 of adulthood are consistent with dysregulation of neuronal gene expression. Refined genetic perturbation within the nervous system coupled with single-cell RNA-Seq over time would be required to fully reveal the impact of HPK-1 function on age-associated changes in gene expression.

We next investigated how many of these genes are predominantly or exclusively expressed in neurons, based on a set of 880 neuron-enriched and neuron-specific genes (*Kaletsky et al., 2018*). We found that 15% of all *hpk-1* upregulated genes are enriched or specific to neurons in wild-type animals (283 genes, hypergeometric test p-value <0.0001) whereas only five *hpk-1* downregulated genes normally exhibit such neuron-restricted expression. To determine what functions are over-represented among the upregulated neuronally expressed genes, we ran GO enrichment analysis. Representing these results as a network, and clustering terms which have overlapping gene associations, we identified some major themes: neurotransmitter and ion transport, synaptic components and signaling, axon and neuron projection components, feeding and locomotion behavior, mechanosensation and touch-responsive behavior are all upregulated in the absence of *hpk-1* (*Figure 4D*). Among all genes upregulated in the *hpk-1* null mutants, the largest group correspond to genes related to neuropeptide encoding, processing, and receptors. Neuropeptides are short sequences of amino acids important for neuromodulation of animal behavior, broadly expressed in sensory, motor, and interneurons. Of the 120 neuropeptide-encoding genes in the *C. elegans* genome, 40 (33%) are upregulated in the *hpk-1* null mutants. Nine of these are insulin-related (*ins*), which suggest increased *hpk-1* expression during normal aging or *hpk-1* overexpression may positively affect longevity through decreased *ins* neuropeptide expression. We surmise that many of the neuronal genes that are induced in the absence of *hpk-1* might be compensatory responses to maintain neuronal function. We conclude that loss of *hpk-1* upregulates expression of a number of neuronal genes including those involved in neuropeptide signaling, neurotransmission, release of synaptic vesicles, calcium homeostasis, and function of sensory neurons (Appendix 2).

## Neuronal HPK-1 regulates proteostasis cell intrinsically and cell non-autonomously via neurotransmission

We hypothesized that HPK-1 maintains neuronal function by fortifying the neural proteome. We measured polyglutamine toxic aggregate formation in neuronal cells (*rgef-1p::Q40::YFP*) of animals overexpressing *hpk-1* pan-neuronally (*rab-3p*). These animals showed a decreased number of fluorescent foci within the nerve ring (*Figure 5Ai–iii, B*). To assess whether this was associated with improved neuronal function, we assessed alterations in locomotion capacity (body bends) after placement in liquid. Increased neuronal expression of *hpk-1* significantly increased the average number of body bends (17.7±1 vs 12.5±1.2 bends/30 s), suggesting that neuronal expression of *hpk-1* improves proteostasis in the nervous system and mitigates locomotion defects associated with the decline in proteostasis (*Figure 5C*).

We next tested whether HPK-1 was necessary to preserve proteostasis within neurons. We crossed the *rgef-1::Q40::YFP* reporter with neuronal enhanced RNAi background (*sid-1(pk3321);unc-119p::sid-1*) and assessed whether neuronal inactivation of *hpk-1* hastened age-associated proteostatic decline. Animals with neuronal *hpk-1* inactivation had a significant increase in foci number within neurons and a decreased number of body bends in liquid (19.7±0.8 vs 25.5±1.1 bends/30 s) (*Figure 5Aiv, D, E*). Thus, HPK-1 functions cell intrinsically to maintain neuronal proteostasis.

We next assessed whether neuronal HPK-1 activates a cell non-autonomous signal to regulate proteostasis in distal tissues. We used a proteostasis reporter expressed exclusively in muscle tissue (*unc-54p::Q35::YFP*); animals display an age-associated accumulation of fluorescent foci, which are proteotoxic aggregates, and become paralyzed as muscle proteome function collapses (*Lazaro-Pena*

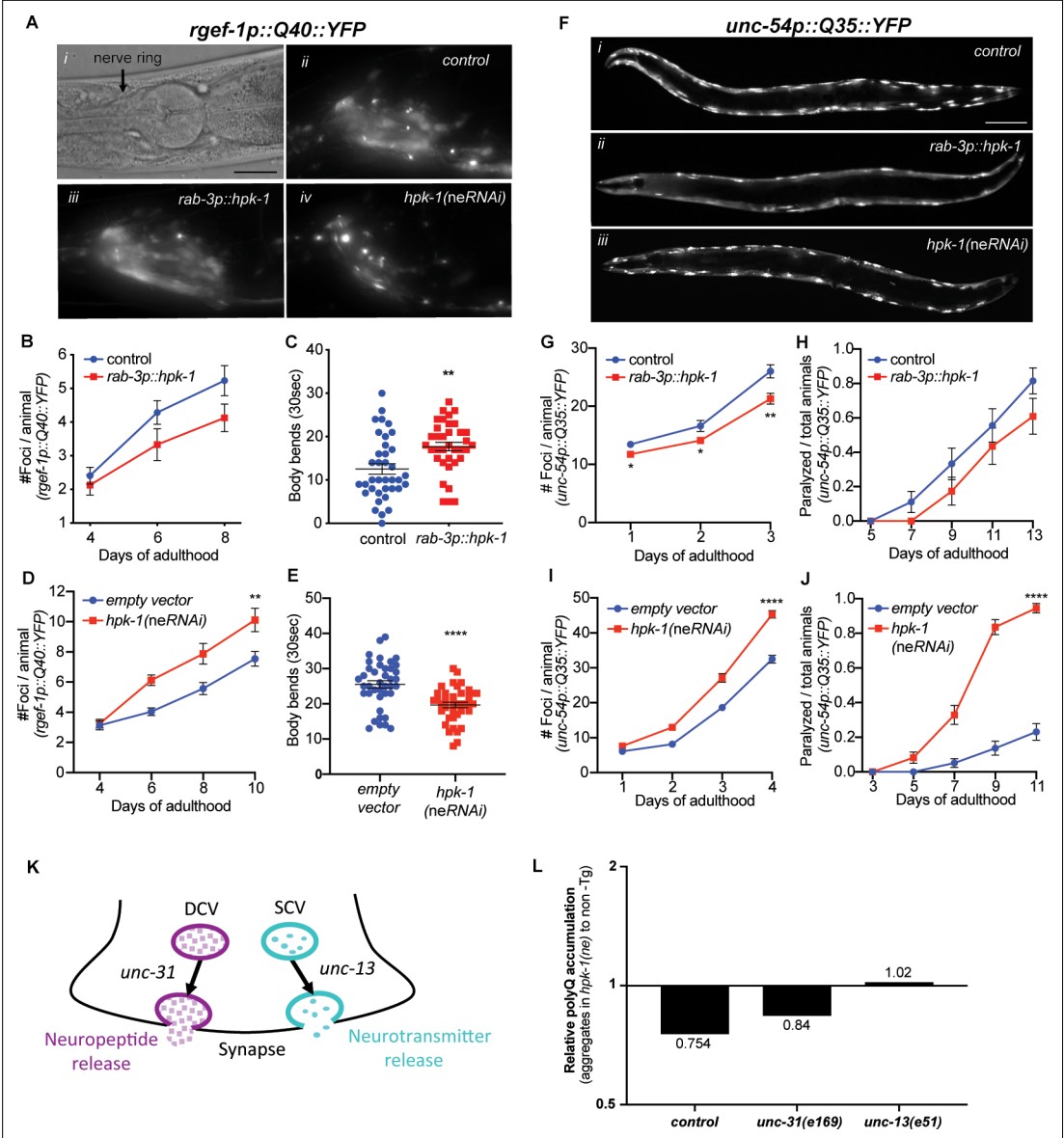

**Figure 5.** Neuronal homeodomain-interacting protein kinase (HPK-1) prevents the decline in proteostasis through neurotransmitter release. (**A**) Nomarski (**i**) and fluorescent (**ii–iv**) micrographs of animals expressing polyglutamine fluorescent reporter within the nervous system. Scale bar, 25 μm. (**B and C**) Quantification of foci in the nerve ring (**B**) and body bend frequency (**C**) (n≥21 for B and n≥36 for **C**). (**D and E**) Quantification of foci in the nerve ring (**D**) and frequency of body bends (**E**) in neuronal enhanced RNAi animals (n≥19 for D and n≥39 for **E**). (**F**) Fluorescent micrographs of animals expressing the polyglutamine fluorescent reporter in muscle. Scale bar, 100 μm. (**G and H**) Quantification of foci in muscle (**G**) and paralysis rate (**H**). Graphs are representative of five individual transgenic lines (n≥18 for G and n≥23 for **H**). (**I and J**) Quantification of foci in muscle (**I**) and paralysis rate (**J**) in neuronal enhanced RNAi animals (n≥22 for I and n≥73 for **J**). (**K**) Neuropeptides and neurotransmitters are essential for neuronal signaling; vesicle release depends on *unc-31* and *unc-13*, respectively. (**L**) Quantification of relative *polyQ* fluorescent foci in muscle cells after *rab-3p::hpk-1* overexpression in control, *unc-31(e169)* and *unc-13(e51)* animals; see *Figure 5—figure supplement 2* for absolute values and statistical analysis. t-Test analysis with **p<0.01 and ***p<0.001. Bars represent ± SEM. See *Supplementary file 2*, *Supplementary file 12* and *Supplementary file 13* for details and additional trials.

The online version of this article includes the following figure supplement(s) for figure 5:

**Figure supplement 1.** Homeodomain-interacting protein kinase (HPK-1) kinase activity is necessary to improve proteostasis.

**Figure supplement 2.** Homeodomain-interacting protein kinase (HPK-1) cell non-autonomous regulation of proteostasis depends on neurotransmitter signaling.

*et al., 2021*; *Morley et al., 2002*). Animals with increased neuronal *hpk-1* expression had significantly decreased fluorescent foci in muscle tissue (*Figure 5Fi–ii*). For instance, at day 3 of adulthood, animals overexpressing neuronal HPK-1 had an average of 21±0.9 polyQ aggregates, while control animals had an average of 26±1.1 polyQ aggregates (*Figure 5G*). In *C. elegans*, the aging-related progressive proteotoxic stress of muscle polyQ expression is pathological and results in paralysis. Of note, increased *hpk-1* expression in neurons was sufficient to reduce paralysis; at day 13 of adulthood only 61%±0.10 of neuronal *hpk-1* overexpressing animals were paralyzed compared to 81%±0.08 of paralyzed control animals (*Figure 5H*). Thus, increased neuronal HPK-1 activity is sufficient to improve proteostasis cell non-autonomously.

We confirmed that the kinase activity is essential for HPK-1 to improve proteostasis. We find the kinase domain of HPK-1 is highly conserved from yeast to mammals, both at the amino acid level and predicted tertiary structure, based on multiple sequence alignments, AlphaFold structure prediction, and visualization with PyMol (*Figure 5—figure supplement 1*; *DeLano, 2002*; *Jumper et al., 2021*; *Varadi et al., 2022*). We mutated two highly conserved residues within the kinase domain, K176A and D272N, which are known to be essential for the kinase activity of HPK-1 orthologs (*He et al., 2010*), and tested whether these kinase-dead mutant versions could improve proteostasis when neuronally overexpressed. In contrast to wild-type HPK-1 (*Figure 5G and H*), neither mutant had any effect on either polyglutamine foci accumulation (*Figure 5—figure supplement 1C, D*) or paralysis (*Figure 5—figure supplement 1E, F*) in muscle cells (*unc-54p::Q35::YFP*); thus, increased *hpk-1* expression improves proteostasis through kinase activity.

We tested whether neuronal HPK-1 is required for maintaining proteostasis in distal muscle cells. Neuronal inactivation of *hpk-1* (*sid-1(pk3321); unc-119p::sid-1; unc-54p::Q35::YFP; hpk-1(RNAi)*) was sufficient to hasten the collapse of proteostasis, significantly increasing the number of polyQ aggregates (45±1 vs 32±1.1 at day 4) and paralysis rate (84%±0.04 vs 14%±0.04 at day 9) (*Figure 5Fiii, I, J*). Thus, HPK-1 function within the nervous system impacts proteostasis in distal muscle tissue, which implies that HPK-1 coordinates organismal health through cell non-autonomous mechanisms.

To identify the signaling mechanism through which HPK-1 regulates distal proteostasis, we tested whether either neuropeptide or neurotransmitter transmission are essential for cell non-autonomous regulation of proteostasis (using *unc-31* and *unc-13* mutant animals, respectively; *Figure 5K*). Neuronal expression of *hpk-1* in *unc-31* mutants improved proteostasis in the absence of neuropeptide transmission and dense core vesicles, as increased neuronal *hpk-1* expression still delayed the age-associated accumulation of aggregates within muscle cells. In contrast, in the absence of neurotransmitter function, neuronal expression of *hpk-1* in *unc-13* mutants failed to improve proteostasis in muscle tissue (*Figure 5—figure supplement 2*). We conclude that HPK-1-mediated longevity-promoting cell non-autonomous signaling occurs through neurotransmission.

## Neuronal HPK-1 induces molecular chaperones and autophagy in distal tissues

We sought to identify the components of the PN that are cell non-autonomously regulated via HPK-1 activity in the nervous system. We tested whether increasing levels of neuronal *hpk-1* alters the expression of molecular chaperones (*hsp-16.2p::GFP*). Increased neuronal expression of *hpk-1* in wild-type animals did not alter the basal expression of molecular chaperones (*Figure 6A*). In contrast, after heat shock, increased neuronal *hpk-1* expression resulted in hyper-induction of *hsp-16.2p::GFP* expression, particularly in the pharynx and head muscles, but surprisingly not within intestinal cells (*Figure 6B–E*). We conclude that while *hpk-1* is required for the broad induction of molecular chaperones (*Das et al., 2017*), neuronal HPK-1 primes inducibility of the HSR within specific tissues, rather than in a systemic manner throughout the organism.

We next tested whether increased neuronal HPK-1 activity is sufficient to induce autophagy. Using two LGG-1/Atg8 fluorescent reporter strains to visualize autophagosome levels in neurons (*rgef-1p::GFP::LGG-1)*, hypodermal seam cells, and intestine (*lgg-1p::GFP::LGG-1*) (*Kumsta et al., 2017*), we found that increased neuronal *hpk-1* expression significantly induces autophagy in neurons and hypodermal seam cell (*Figure 6F–I*), but not in the intestine (*Figure 6J and K*). Sustained autophagy and transcriptional activation of autophagy genes are important components for somatic maintenance and organismal longevity (*Lapierre et al., 2015*). Consistently, *hpk-1* is required for the induction of autophagy gene expression induced by TORC1 inactivation (*Das et al., 2017*). We tested whether

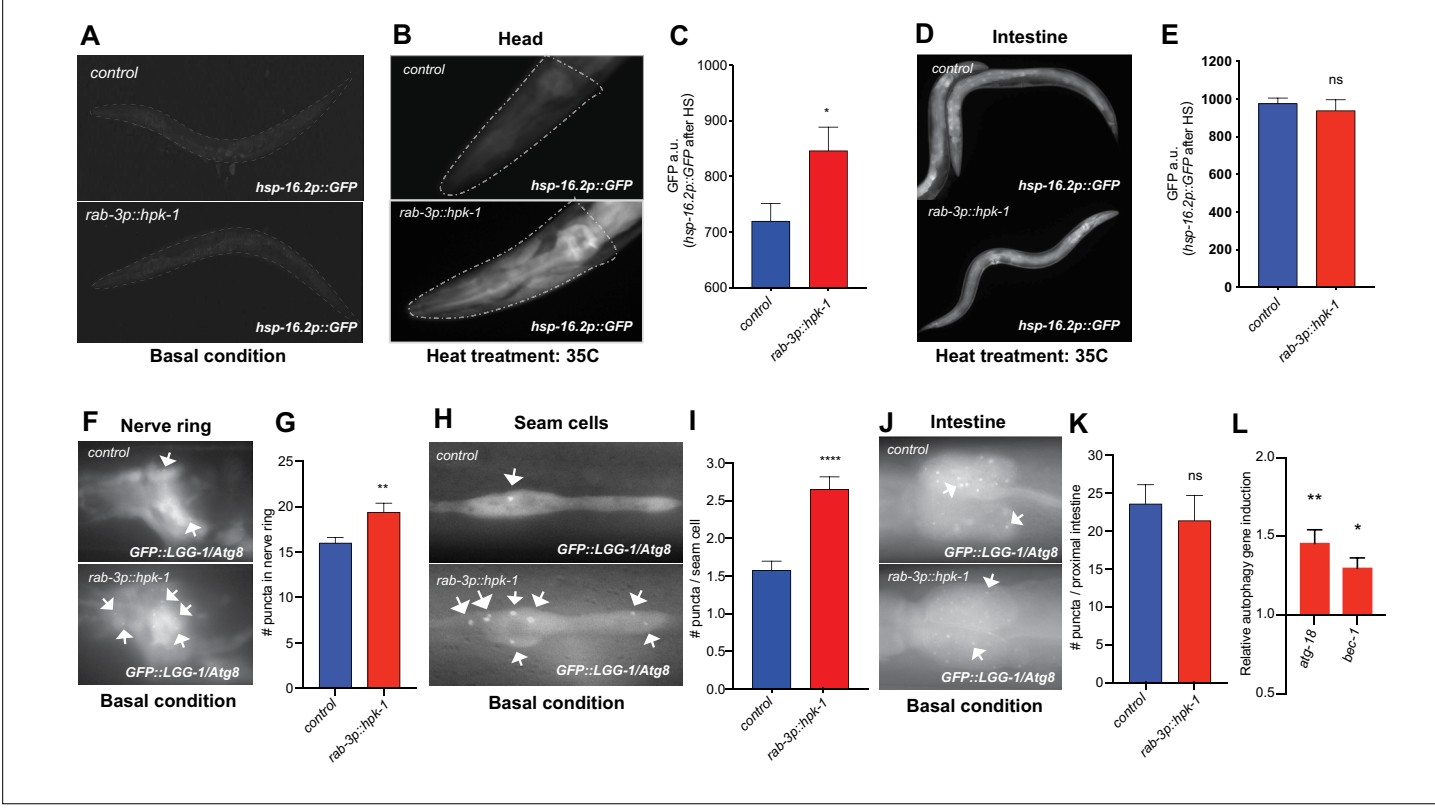

**Figure 6.** Neuronal homeodomain-interacting protein kinase (HPK-1) induces the proteostatic network in distal tissues. (**A**) Fluorescent micrographs of animals expressing *hsp-16.2p::GFP* in basal conditions. (**B and C**) Fluorescent micrographs (**B**) and fluorescent density quantification (**C**) of control and *rab-3p::hpk-1* day 1 adult animals expressing *hsp-16.2p::GFP* after heat shock (35°C) for 1 hr at (n≥15). (**D and E**) Fluorescent micrographs and densitometry quantification of the intestinal fluorescence of *hsp-16.2p::GFP* after heat shock (n≥24). (**F and G**) Fluorescent micrographs and quantification of fluorescent puncta in the nerve ring of control and *rab-3p::hpk-1* L4 animals expressing *rgef-1p::GFP::LGG-1/Atg8* (n≥21). (**H and I**) Fluorescent micrographs (**H**) and quantification (**I**) of puncta in seam cells of control and *rab-3p::hpk-1* animals expressing *lgg-1p::GFP::LGG-1/Atg8* (n≥43). (**J and K**) Fluorescent micrographs (**J**) and quantification (**K**) of puncta in proximal intestinal cells of control and *rab-3p::hpk-1* animals expressing the *lgg-1p::GFP::LGG-1/Atg8* autophagosome reporter (n≥17). (**L**) Expression of autophagy genes *atg-18* and *bec-1* via RT-qPCR in *rab-3p::HPK-1* animals compared to non-transgenic controls. t-Test analysis with *p<0.05, **p<0.01, ***p<0.001, and ****p<0.0001. Bars represent ± SEM. See *Supplementary file 4*, *Supplementary file 14* and *Supplementary file 15* for details and additional trials.

neuronal overexpression of *hpk-1* is sufficient to induce autophagy gene expression, and found that increased neuronal overexpression induces expression of *atg-18* (WD repeat domain, phosphoinositide interacting 2) and *bec-1* (Beclin 1) (*Figure 6L*). Thus, increased HPK-1 activity within the nervous system is sufficient to induce basal levels of autophagy both cell intrinsically and non-autonomously, which we posit occurs through increased autophagy gene expression to extend longevity.

## GABAergic and serotonergic HPK-1 differentially regulate longevity and acute stress signaling

*hpk-1* is broadly expressed within the *C. elegans* adult nervous system (*Das et al., 2017*), but the neuronal cell types responsible for extending longevity are unknown. We overexpressed *hpk-1* in different subtypes of neurons and assessed changes in muscle proteostasis (*unc-54p::Q35::YFP*). Increased *hpk-1* expression in glutamatergic or dopaminergic neurons was not sufficient to decrease the formation of fluorescent foci or reduce paralysis (*Figure 7—figure supplement 1*). While it is possible that *hpk-1* overexpression in these neuronal subtypes failed to reach a threshold effect, these results suggest that HPK-1 does not act within these neuronal subtypes to cell non-autonomously regulate proteostasis. In contrast, increased expression of *hpk-1* in either serotonergic (*tph-1p::hpk-1*) or GABAergic (*unc-47p::hpk-1*) neurons was sufficient to improve proteostasis in muscle tissue (*Figure 7A–D*). Accordingly, we found that *hpk-1* is expressed in serotonergic and GABAergic

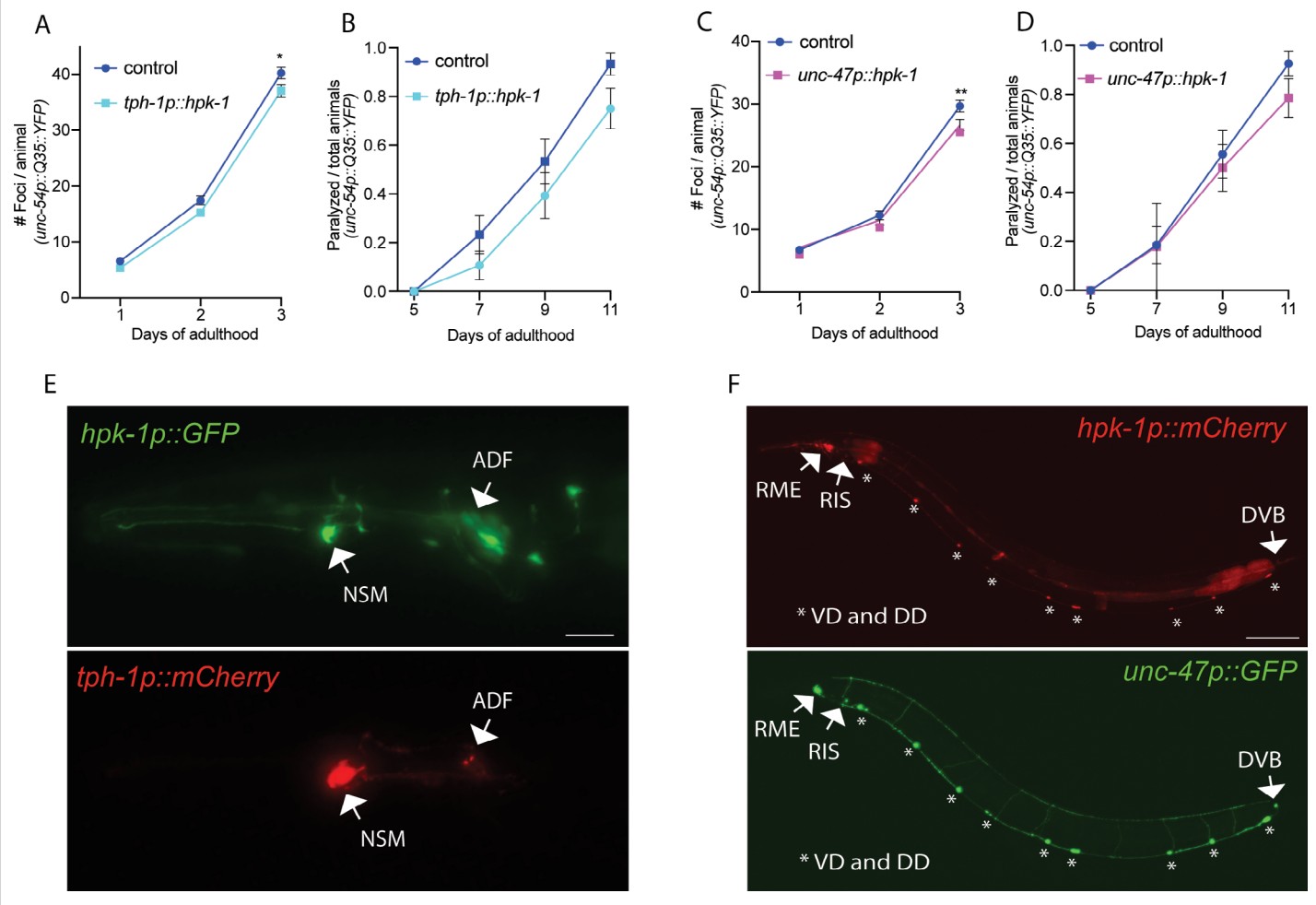

**Figure 7.** Expression of homeodomain-interacting protein kinase (HPK-1) in serotonergic and γ-aminobutyric acid (GABA)ergic prevents the decline in proteostasis in muscle tissue. (**A and B**) Quantification of foci (**A**) and paralysis rate (**B**) of animals expressing *hpk-1* in serotonergic neurons (n≥17 for A and n≥28 for B). (**C and D**) Quantification of foci (**C**) and paralysis rate (**D**) of animals expressing *hpk-1* in GABAergic neurons (n≥19 for **C** and n≥27 for **D**). (**E**) Representative fluorescent micrographs of *hpk-1* expression in serotonergic neurons. Scale bar, 25 μm. (**F**) Representative fluorescent micrographs of *hpk-1* expression in GABAergic neurons. Scale bar, 100 μm. t-Test analysis performed to each time point with *p<0.05 and **p<0.01. Bars represent ± SEM. See *Supplementary file 12* and *Supplementary file 13* for details and additional trials.

The online version of this article includes the following figure supplement(s) for figure 7:

**Figure supplement 1.** *Hpk-1* expression in glutamatergic and dopaminergic neurons is not sufficient to prevent the decline in proteostasis.

**Figure supplement 2.** *Hpk-1* is expressed in γ-aminobutyric acid (GABA)ergic and serotonergic neurons.

neurons, based on colocalization between *hpk-1* and neuronal cell-type specific reporters (*Figure 7E and F*, *Figure 7—figure supplement 2*), consistent with the notion that HPK-1 initiates pro-longevity signals from these neurons.

We sought to determine whether signals initiated from serotonergic and GABAergic neurons induced similar or distinct components of the PN in distal tissues, and assessed the overall consequence on organismal health. Since pan-neuronal expression of *hpk-1* was sufficient to induce autophagy, improve thermotolerance, and increase lifespan, we chose to dissect whether these three phenotypes were regulated within distinct neuronal cell types. Increased HPK-1 serotonergic signaling was sufficient to improve thermotolerance, without altering lifespan or autophagy (*Figure 8A–D*). In contrast, increased *hpk-1* expression in GABAergic neurons did not alter thermotolerance, but induced autophagy slightly but significantly increased lifespan (*Figure 8E–H*).

To identify the mechanism through which increased neuronal HPK-1 activity induces autophagy in distal tissues, we examined serotonergic neurons. As expected, loss of *tph-1*, which encodes

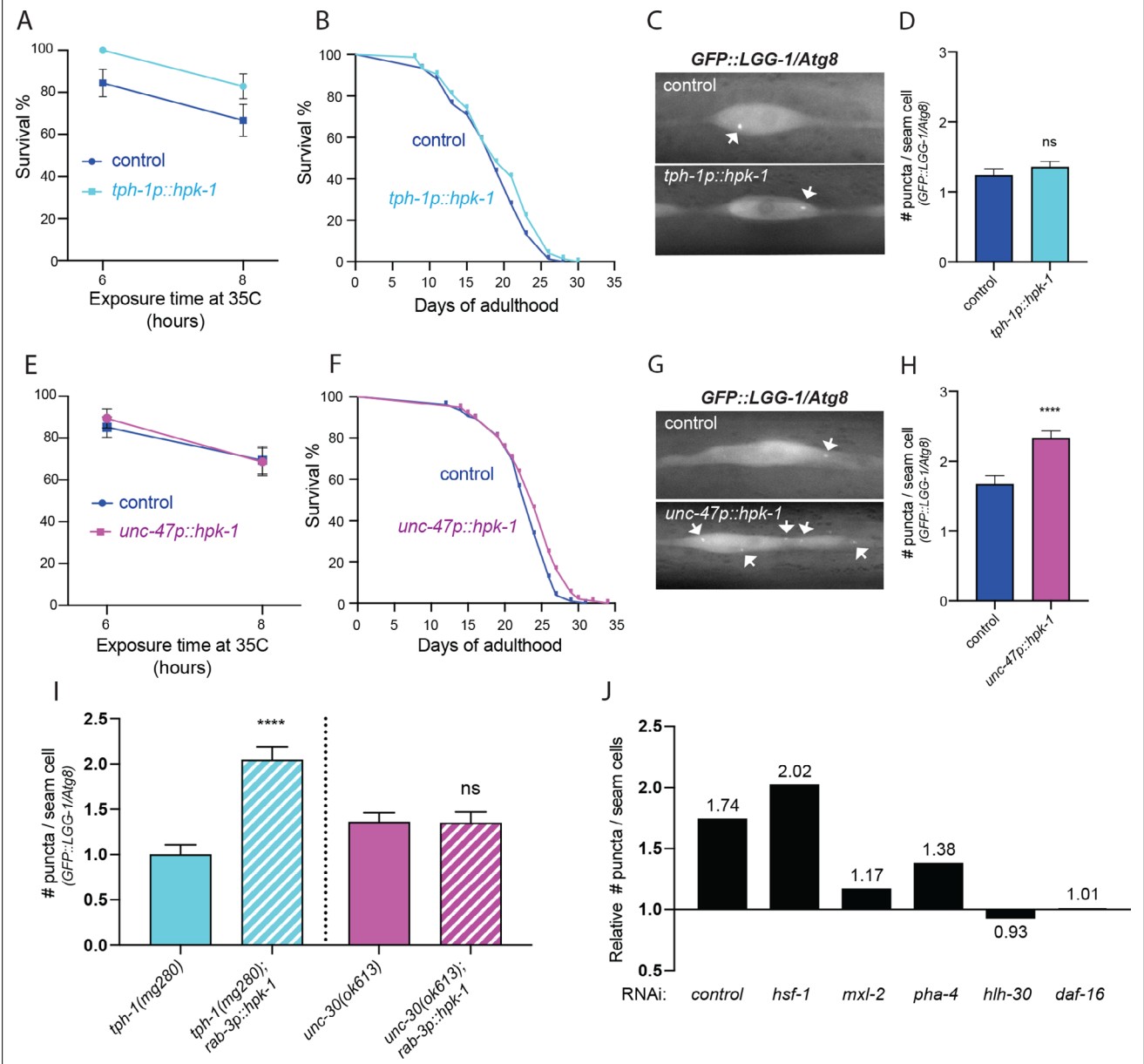

**Figure 8.** Serotonergic and γ-aminobutyric acid (GABA)ergic homeodomain-interacting protein kinase (HPK-1) signaling activates distinct adaptive responses to regulate stress resistance and longevity. (**A**) Survival of serotonergic HPK-1 expressing day 1 adult animals after heat shock (n≥32). (**B**) Lifespan of serotonergic HPK-1 expressing animals (n≥76). (**C and D**) Representative fluorescent micrographs (**C**) and quantification (**D**) of autophagosomes in hypodermal cells of serotonergic HPK-1 day 1 adult animals (n≥80). (**E**) Survival of GABAergic HPK-1 expressing day 1 adult animals after heat shock (n≥47). (**F**) Lifespan of GABAergic HPK-1 expressing animals (n≥119). (**G and H**) Representative fluorescent micrographs (**G**) and quantification of autophagosomes in hypodermal cells of GABAergic HPK-1 day 1 adult animals (n≥64). (**I**) Quantification of autophagosomes in hypodermal cells of mutant animals of serotonin and GABAergic signaling (*tph-1* and *unc-30*, respectively), expressing pan-neuronal *hpk-1* (day 1 adults, n≥67). (**J**) Relative change in number of puncta in seam cells of *unc-47p::hpk-1* animals expressing *lgg-1p::GFP::LGG-1/Atg8*, compared to non-transgenic controls after indicated gene inactivation (n≥28) for all conditions; see *Figure 8—figure supplement 1* for absolute values and statistical analysis. t-Test analysis with *p<0.05, **p<0.01, and ****p<0.0001. Bars represent ± SEM. See *Supplementary file 1*, *Supplementary file 3*, and *Supplementary file 15* for details and additional trials.

The online version of this article includes the following figure supplement(s) for figure 8:

**Figure supplement 1.** HLH-30 (TFEB), MXL-2 (Mlx), and DAF-16 (FoxO) are required for homeodomain-interacting protein kinase (HPK-1)-mediated induction of autophagy.

**Figure supplement 2.** *mxl-2* is required for decreased target of rapamycin complex 1 (TORC1) signaling to increase longevity.

the *C. elegans* tryptophan hydroxylase 1 essential for biosynthesis of serotonin (*Sze et al., 2000*), failed to block the induction of autophagy by increased neuronal *hpk-1* expression (*Figure 8I*, *tph-1(mg280);rab-3p::hpk-1*). *unc-30* encodes the ortholog of human paired like homeodomain 2 (PITX2) and is essential for the proper differentiation of type-D inhibitory GABAergic motor neurons (*Jin et al., 1994*; *McIntire et al., 1993*). Consistent with our previous result demonstrating that only GABAergic expression of *hpk-1* promotes autophagy, loss of *unc-30* was sufficient to completely abrogate the induction of autophagy in seam cells after increased pan-neuronal expression of *hpk-1* (*Figure 8I*, *unc-30(ok613);rab-3p::hpk-1*). We conclude that HPK-1 activity in serotonergic neurons promotes systemic thermotolerance, whereas HPK-1 in GABAergic neurons promotes autophagy. We surmise that the two neuronal subtypes are likely to contribute synergistically to the extension of longevity.

HPK-1 regulates autophagy gene expression and across diverse species HIPK family members directly regulate TFs (*Sardina et al., 2023*), therefore we sought to identify TFs essential for HPK-1 overexpression within GABAergic neurons to induce autophagy. *pha-4* (FOXA), *mxl-2* (MLX), *hlh-30* (TFEB), *daf-16* (FOXO), and *hsf-1* are all required in specific contexts for the induction of autophagy and longevity (*Hansen et al., 2008*; *Kumsta et al., 2017*; *Lapierre et al., 2013*; *Lapierre et al., 2015*; *Nakamura et al., 2016*; *O'Rourke and Ruvkun, 2013*). Furthermore, we previously found that *hsf-1, mxl-2, or pha-4* are all necessary for the increased lifespan conferred by *hpk-1* overexpression throughout the soma (*Das et al., 2017*). Using feeding-based RNAi, we found that *hpk-1* overexpression in GABAergic neurons still significantly induced autophagosomes in seam cells after loss of *hsf-1* or *pha-4* (*Figure 8J*); thus HPK-1 induction of autophagy is independent of *hsf-1* and *pha-4* and suggests the *pha-4* requirement for HPK-1-mediated longevity is independent of autophagy gene regulation (at least in hypodermal seam cells). In contrast, after inactivation of *mxl-2, hlh-30,* or *daf-16,* GABAergic overexpression of *hpk-1* failed to significantly increase autophagosomes (*Figure 8J*); thus HPK-1-mediated induction of autophagy requires *mxl-2, hlh-30*, and *daf-16*. We were puzzled to find that the average number of autophagosome in control animals (i.e., lacking *hpk-1* overexpression) varied after gene inactivation (*Figure 8—figure supplement 1A*). To rigorously test whether *hpk-1* overexpression required *hlh-30* for the induction of autophagosomes, we crossed the *hlh-30* null mutation into animals overexpressing *hpk-1* within GABAergic neurons (*unc-47p::hpk-1;hlh-30(tm1978);lgg-1p::GFP::LGG-1*); we find that *hlh-30* is required for the induction of autophagosomes in seam cells when *hpk-1* is overexpressed in GABAergic neurons (*Figure 8—figure supplement 1B*).

We sought to fortify our previous findings that: (1) TORC1 negatively regulates *hpk-1* under basal conditions; (2) *hpk-1* is essential for decreased TORC1 signaling to increase longevity and induce autophagy; and (3) HPK-1-mediated longevity requires *mxl-2*, the ortholog of mammalian Mlx TF (*Das et al., 2017*). From these data, we predicted that TOR-mediated longevity might require *mxl-2* via *hpk-1*. Indeed, we find that inactivation of either *daf-15* (Raptor) or *let-363* (TOR) increased lifespan and required *mxl-2*, as *mxl-2(tm1516)* null mutant animals failed to increase lifespan in the absence of TORC1 (*Figure 8—figure supplement 2*). MXL-2 functions as a heterodimer with MML-1 (*Pickett et al., 2007*), which have similar roles in *C. elegans* longevity (*Johnson et al., 2014*). Interestingly, one of the human homologs of MML-1, carbohydrate response element binding protein (ChREBP/MondoB/MLXIPL/WBSCR14), maps within a deleted region in Williams-Beuren syndrome patients, which among other symptoms causes metabolic dysfunction, silent diabetes, and premature aging (*Pober, 2010*). This further supports our model and is consistent with other studies linking the Myc-family of TFs to longevity, autophagy, and TORC1 signaling (*Johnson et al., 2014*; *Nakamura et al., 2016*; *Samuelson et al., 2007*). Collectively, we conclude that HPK-1 regulates serotonergic signals to improve survival in acute thermal stress, likely through HSF-1, and initiates GABAergic signals to extend longevity through increased autophagy, which is linked to decreased TORC1 signaling and regulation of the Myc TFs.

## Discussion

It has been known that a core group of TFs integrates various types of metabolic and stress signals; moreover, this group of TFs can extend longevity when overexpressed, and conversely hasten aging when lost (reviewed in *Denzel et al., 2019*), but whether the activity or expression of these longevity-associated transcriptional regulators are activated by signals of increasing damage or dysfunction during normal aging was unknown. Recently, single-cell RNA-Seq has revealed that these longevity-associated TFs increase in expression during aging in wild-type animals across multiple somatic tissues

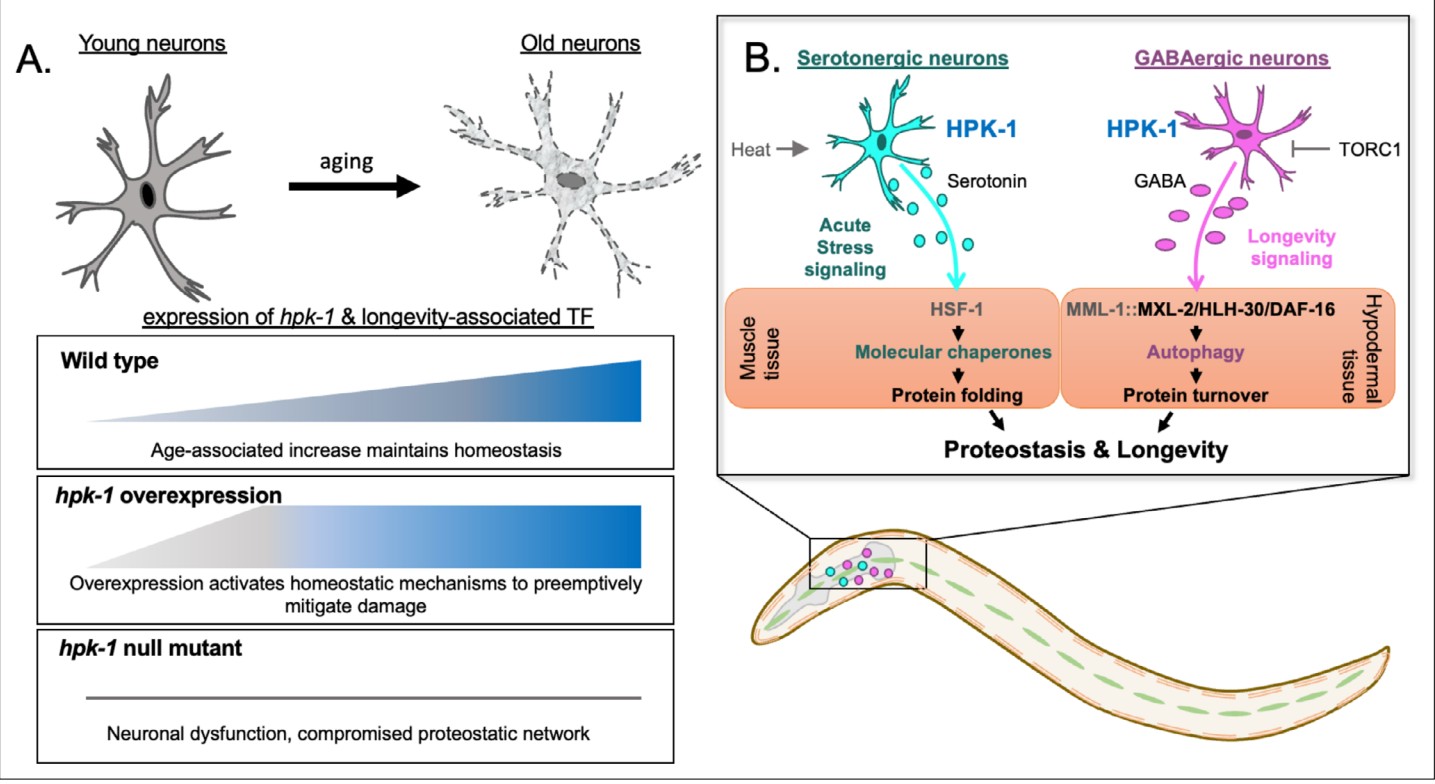

**Figure 9.** Differential regulation of the proteostatic network through homeodomain-interacting protein kinase (HPK-1) activity in serotonergic and γ-aminobutyric acid (GABA)ergic neurons. (**A**) During normal aging *hpk-1* expression increases broadly throughout the nervous system in response to accumulating damage and dysfunction. Overexpression of *hpk-1* within the nervous system fortifies proteostasis by priming and preemptively activating mechanisms that delay the progression of aging. In the absence of *hpk-1* the proteostatic network is compromised, resulting in neuronal dysfunction and increased expression compensatory mechanisms to maintain homeostasis (see text for details). (**B**) HPK-1 activity in serotonergic and GABAergic neurons initiates distinct adaptive responses, either of which improve proteostasis in a cell non-autonomous manner. Serotonergic HPK-1 protects the proteome from acute heat stress, while GABAergic HPK-1 fortifies the proteome by regulating autophagy activity in response to metabolic stress, such as changes in target of rapamycin complex 1 (TORC1) activity. The MXL-2, HLH-30 (TFEB), and DAF-16 (FOXO) transcription factors are all required for hypodermal induction of autophagy in response to increased HPK-1 activity in GABAergic neurons. Gray indicates predicted interactions based on our prior genetic analysis (*Das et al., 2017*). MML-1 and MXL-2 heterodimerize and encode the orthologs of the Myc-family of transcription factors Mondo A/ChREBP and Mlx, respectively.

(*Roux et al., 2022*). This comprehensive empirical dataset provides foundational support for a key theme in aging research: during aging in wild-type animals, increasing damage and dysfunction is mitigated by the activation of stress response programs. Our findings that: (1) *hpk-1* expression is upregulated during normal aging in more neurons and non-neuronal cells than any other kinase, and (2) that *hpk-1* is induced in cell clusters that overlap with longevity-associated TFs, including *daf-16, hlh-30, skn-1*, and *hif-1*, implies that HPK-1 is a key transcriptional regulator that mitigates aging. We posit this response is activated in the nervous system during normal aging in wild-type animals to mitigate increasing stress and the breakdown of homeostatic mechanisms within the nervous system. Consistently, the presence of human neurodegenerative disorders, a severe form of neuronal proteotoxic stress, correlates with activation of human *hpk-1* expression: for example, human HIPK2 is activated in frontal cortex samples of patients with AD, in motor neurons of ALS patients, and in mouse models of ALS (*Lee et al., 2016*). Thus, our results suggest that overexpression of *hpk-1* in the *C. elegans* nervous system may extend longevity by preemptively delaying the accumulation of age-associated damage and dysfunction to the proteome (*Figure 9A*). However, *hpk-1* homologs in more complex metazoans can also promote cell death, a key homeostatic component of tissue regeneration, indicating that chronic activation of HIPKs is not always beneficial (*Calzado et al., 2007*; *Lee et al., 2016*; *Wang et al., 2020*; *Zhang et al., 2020*). Whether activated HIPKs will fortify proteostasis and extend longevity, or promote cell death is likely to be dependent on many factors, including the specific homolog, levels of expression, physiological context, genetic background, and cell type.

We find that *hpk-1* prevents age-associated decline of axonal and synaptic integrity, which together preserve neuronal function during the aging process and thus also promote longer healthspan. In support of this concept, adult *hipk2-/-* mice display normal neurogenesis but manifest delayed-onset severe psychomotor and behavioral deficiencies, as well as degeneration of dopaminergic neurons, reminiscent of PD (*Zhang et al., 2007*). Loss of *hpk-1* results in dysregulation of neuronal gene expression by early adulthood; some changes are consistent with neuronal aging but in other cases we observe an inverse correlation. For example, one-third of all neuropeptides are induced in the absence of *hpk-1*; yet increased neuropeptide signaling has been associated with extended *C. elegans* longevity and is considered protective in AD (*Chen et al., 2018*; *Hodge et al., 2022*; *Petrella et al., 2019*). One possibility is in the absence of *hpk-1*, attenuated inducibility of the proteostatic network accelerates age-associated cellular dysfunction (*Figure 9A*), and upregulation of neuropeptide signaling compensates in the nervous system for the loss of proteostatic response. This concept is consistent with our finding that increasing neuronal HPK-1 activity does not require neuropeptides to improve proteostasis in distal tissues. An alternative possibility for the connection between protective effects of HPK-1 and neuropeptides in neurons is that HPK-1 may reduce insulin signaling by suppressing the processing of *ins* peptides, thereby decreasing ILS to activate DAF-16 (FOXO) transcriptional quality control programs. For example, loss of *egl-3*, a neuropeptide processing gene, extends longevity by inactivating processing of insulins (*Hamilton et al., 2005*), and *egl-3* is upregulated in *hpk-1* loss of function.

Impairment of the proteostatic network after loss of *hpk-1* results in broad upregulation of gene expression, which may provide additional levels of compensatory mechanisms to mitigate neuronal dysfunction. For example, loss of *hpk-1* induces expression of intraflagellar transport (IFT) components in sensory cilia. Mutations within sensory neurons can alter longevity (*Alcedo and Kenyon, 2004*; *Apfeld and Kenyon, 1999*), and decreased IFT during *C. elegans* aging impairs sensory perception and metabolism (*Zhang et al., 2021*). Deficiencies in primary cilia impairs neurogenesis in AD transgenic mice, resulting in accumulation of A$\beta_{42}$ and tau protein and disease progression (*Armato et al., 2013*; *Chakravarthy et al., 2012*; *Ma et al., 2022*; *Morelli et al., 2017*). However, our bulk RNA-Seq analysis cannot decipher changes in gene expression at the level of individual neurons after loss of *hpk-1*. *hpk-1* and orthologs are known to have roles in development, differentiation, and cell fate specification (*Berber et al., 2013*; *Blaquiere and Verheyen, 2017*; *Calzado et al., 2007*; *Hattangadi et al., 2010*; *Rinaldo et al., 2007*; *Rinaldo et al., 2008*; *Steinmetz et al., 2018*), therefore we cannot preclude the possibility that HPK-1 may function to broadly limit neuronal gene expression during the course of normal development and differentiation, thereby specifying neuronal transcriptional programs. Mis-differentiation of the nervous system in the absence of *hpk-1* could result in both intracellular transcriptional noise and cell non-autonomous noise across cell types and tissues.

We show that HPK-1 expression in serotonergic neurons activates a cell non-autonomous signal to prime induction of molecular chaperones in muscle cells, whereas HPK-1 expression in GABAergic neurons signals induces autophagy in hypodermal tissue (*Figure 9B*). Our findings place HPK-1 within a selective cohort of transcriptional regulators within the *C. elegans* nervous system that are activated in response to metabolic, environmental, and intrinsic cues; thereby initiating distinct cell non-autonomous signals to regulate components of the PN in peripheral tissues (*Douglas et al., 2015*; *Durieux et al., 2011*; *Frakes et al., 2020*; *Imanikia et al., 2019*; *Prahlad et al., 2008*; *Prahlad and Morimoto, 2011*; *Silva et al., 2013*; *Tatum et al., 2015*; *Taylor and Dillin, 2013*; *Morimoto, 2020*). Loss of serotonin production in *tph-1* mutant animals has a minimal impact on lifespan at basal temperature (*Murakami and Murakami, 2007*). However, simultaneous loss of both serotonin and DA biosynthesis shortens lifespan (*Murakami and Murakami, 2007*). Serotonin biosynthesis is also required for several conditions that extend longevity: animals treated with an atypical antidepressant serotonin antagonist (mianserin) (*Petrascheck et al., 2007*), changes in food perception (*Entchev et al., 2015*), and neuronal cell non-autonomous induction of the mitochondrial unfolded protein response (*Berendzen et al., 2016*; *Zhang et al., 2018*). In contrast, loss of either of the serotonin receptors *ser-1* or *ser-4* have opposing effects on lifespan, suggesting antagonistic activity between serotonin receptors (*Murakami and Murakami, 2007*).

Surprisingly, we find that increased HPK-1 activity in serotonergic neurons does not increase lifespan, but is sufficient to improve muscle proteostasis and amplify the expression of molecular chaperones that maintain proteostasis within the cytosol and nucleus. Interestingly, neuronal HSF-1

regulation of the HSR through thermosensory circuits is also separable from lifespan (*Douglas et al., 2015*). Serotonin signaling and the serotonin receptor *ser-1* are essential for the induction of the HSR (*Prahlad et al., 2008*; *Tatum et al., 2015*). ADF serotoninergic neurons receive upstream stress signals from the thermosensory circuit, consisting of the amphid sensory neurons (AFD) and the AIY interneurons, which regulate HSF-1 activity in the germline in a serotonin-dependent manner (*Tatum et al., 2015*). However, increased serotonergic HPK-1 activity is not equivalent to altered thermosensory signals: *hpk-1* improves proteostasis within body wall muscles but not intestinal cells through neurotransmitters, and is independent of dense core vesicles. In contrast, loss of the guanylyl cyclase (*gcy-8*) in AFD thermosensory neurons or the LIM-homeobox TF (*ttx-3*) in AIY interneurons improves intestinal proteostasis through dense core vesicle neurosecretion (i.e., *unc-31*). *hpk-1* is expressed in AFD, but only at very low levels in AIY neurons (data not shown and *Taylor et al., 2021*), and *hpk-1* is essential for the induction of the HSR within intestinal cells (*Das et al., 2017*). Whether *hpk-1* activity within thermosensory neurons functions within the canonical thermosensory-serotonin circuit to regulate the intestinal HSR or proteostasis is not mutually exclusive from our findings and will be interesting to pursue in future studies. Our results suggest that serotonergic HPK-1 is an important regulator of the HSR, potentially functioning in parallel to the thermosensory-serotonin circuit to restore proteostasis in response to acute thermal stress.

We posit that increased serotonergic HPK-1 activity further activates HSF-1 in response to heat (*Figure 9B*). This would be consistent with known hyperphosphorylation of HSF-1 and amplification of the HSR in response to increasingly severe heat stress (reviewed in *Lazaro-Pena et al., 2022*). Interestingly, HPK-1 limits SUMOylation of HSF-1, SUMO limits the HSR in both *C. elegans* and mammals, and severe heat shock results in HSF1 deSUMOylation (*Das et al., 2017*; *Hietakangas et al., 2003*; *Hietakangas et al., 2006*). To our surprise, while increased neuronal *hsf-1* expression increases longevity (*Douglas et al., 2015*; *Morley and Morimoto, 2004*), we could find no study that has identified specific neuronal cell types in which increased HSF-1 activity is sufficient to extend longevity or enhance the HSR. Whether or not HPK-1 regulates HSF-1 within the nervous system and/or in peripheral muscle cells through cell non-autonomous signals will be interesting to determine in future studies. HPK-1 co-localizes with HSF-1 in neurons, HPK-1 protein is stabilized after heat shock, *hpk-1* is essential for induction of the HSR, and *hsf-1* and *hpk-1* are mutually interdependent to extend longevity (*Das et al., 2017*; *Figure 9B*).

Increased HPK-1 activity in GABAergic neurons induces autophagy in peripheral cells, improves proteostasis and longevity. Whether HPK-1 functions in GABAergic neurons through alteration of GABA or another neurotransmitter is unknown, but proper differentiation of GABAergic neurons is essential for neuronal HPK-1 activation of autophagy in distal tissue. We posit that HPK-1 cell non-autonomous signaling occurs via antagonism of GABA signaling. Loss of GABA increases *C. elegans* lifespan (*Chun et al., 2015*; *Yuan et al., 2019*). However, not all neurotransmitters regulate lifespan, as only mutants for the GABA biosynthetic enzyme, *unc-25*, increase lifespan. This GABAergic-mediated control in lifespan seems to be mediated by the *gbb-1* metabotropic GABA_B receptor (*Chun et al., 2015*). Interestingly, GABA signaling inhibits autophagy and causes oxidative stress in mice, which is mitigated via Tor1 inhibition (*Lakhani et al., 2014*). In fact, we previously found that neuronal expression of HPK-1 is inhibited by TORC1 and *hpk-1* is required for TORC1 inactivation to extend longevity and the induction of autophagy gene expression (*Das et al., 2017*). Both TORC1 inactivation and *hpk-1* overexpression requires the Myc-family member *mxl-2* to extend longevity (*Das et al., 2017*; *Johnson et al., 2014*; *Nakamura et al., 2016*; *Figure 8—figure supplement 2*). Consistently, TORC1 also acts within neurons to regulate aging (*Zhang et al., 2019*); our results suggest that GABAergic neurons might be the specific neuronal cell type through which alterations in TORC1 activity regulate longevity. Collectively, our results position HPK-1 activity in GABAergic neurons as the likely integration point for cell non-autonomous regulation of autophagy via TORC1 (*Figure 9B*).

A growing number of longevity-associated TFs have been linked to autophagy, including *pha-4* (FOXA), *mxl-2* (MLX), *hlh-30* (TFEB), *daf-16* (FOXO), and *hsf-1* (*Hansen et al., 2008*; *Kumsta et al., 2017*; *Lapierre et al., 2013*; *Lapierre et al., 2015*; *Nakamura et al., 2016*; *O'Rourke and Ruvkun, 2013*). We find that *hlh-30*, *daf-16*, and *hlh-30* are all required for the induction of autophagosomes

when *hpk-1* is overexpressed in GABAergic neurons. We posit these TFs respond within the distal hypodermal seam cells in response to a neuronal signal initiated by HPK-1, as neurons are largely refractory to RNAi. However, this does not preclude cell intrinsic neuronal interactions; future refined spatial genetic analysis will be required to definitively decipher the precise tissues of action. DAF-16 and HLH-30 have previously been shown to function in combination to promote stress resistance and longevity (*Lin et al., 2018*). Analogously, the MML-1::MXL-2 complex and HLH-30 activate each other and are downstream effectors of longevity after TORC1 inhibition (*mml-1* encodes the MondoA/ ChREBP ortholog, an obligate interactor with MXL-2) (*Nakamura et al., 2016*). Based on our discovery that all three of these longevity-associated TFs are essential for GABAergic HPK-1 activity to induce autophagy, it is tempting to speculate HPK-1 acts as a key mediator of pro-longevity signaling after TORC1 inhibition through coordinated activation of MML-1::MXL-2, HLH-30, and DAF-16 (*Figure 9B*).

This work reveals a new level of specificity in the capacity of the nervous system to initiate adaptive responses by triggering distinct components of the proteostatic network across cell types. Results elucidating how neuronal cell types generate heterotypic signals to coordinate the maintenance of homeostasis are just beginning to emerge. For example, a previous study discovered that activating non-autonomous signaling of the ER-UPR via the XBP-1 TF can be initiated from serotonergic neurons and dopaminergic neurons; these signals are unique and converge to activate distinct branches of the ER-UPR within the same distal tissues (intestinal cells) (*Higuchi-Sanabria et al., 2020*). In contrast, our work demonstrates differential, yet specific, activation of cytosolic components of the PN in separate peripheral tissues through neurotransmitters. Our work suggests that the nervous system partitions neuronal cell types to coordinate the activation of complementary proteostatic mechanisms across tissues, despite utilizing the same transcriptional regulator within each neuronal cell type. One possibility is that HPK-1 regulates specific adaptive responses based on the unique epigenetic states between differentiated neuronal subtypes, which would be consistent with recent findings: during development and differentiation, transcriptional rewiring of cells sets chaperoning capacities and alters usage of mechanisms that maintain protein quality control (*Nisaa and Ben-Zvi, 2022*; *Shemesh et al., 2021*). Our work reveals HPK-1 as a novel transcriptional regulator within the nervous system of metazoans, which integrates diverse stimuli to coordinate specific adaptive responses that promote healthy aging.

## Materials and methods

**Key resources table**

| Reagent type (species) or resource | Designation | Source or reference | Identifiers | Additional information |
|---|---|---|---|---|
| Genetic reagent (*C. elegans*) | N2 | N2 (CGCM) | | *Wild type* |
| Genetic reagent (*C. elegans*) | AVS392 | EK273 | | *hpk-1(pk1393) X* |
| Genetic reagent (*C. elegans*) | AVS543 | This paper | *artEx41* | *artEx41 [rab-3p::hpk-1::CFP+pCFJ90 (myo-2p::mCherry)]* |
| Genetic reagent (*C. elegans*) | AVS609 | This paper | *artEx43* | *artEx43 [myo-3p::hpk-1+pCFJ90 (myo-2p::mCherry)] line 1* |
| Genetic reagent (*C. elegans*) | AVS614 | This paper | *artEx48* | *artEx48 [dpy-7p::hpk-1+pCFJ90 (myo-2p::mCherry)] line 1* |
| Genetic reagent (*C. elegans*) | AVS627 | This paper | *artEx52* | *artEx52 [ges-1p::hpk-1+pCFJ90 (myo-2p::mCherry)] line 1* |
| Genetic reagent (*C. elegans*) | AVS420 | This paper | *artEx41* | *hpk-1(pk1393) X; artEx41 [rab-3p::hpk-1::CFP+pCFJ90 (myo-2p::mCherry)]* |
| Genetic reagent (*C. elegans*) | AVS602 | CZ13799 | *juIs76* | *juIs76 [unc-25p::GFP+lin-15(+)] II* |
| Genetic reagent (*C. elegans*) | AVS608 | This paper | *juIs76* | *hpk-1(pk1393); juIs76 [unc-25p::GFP+lin-15(+)] II* |

*Continued on next page*

Continued

| Reagent type (species) or resource | Designation | Source or reference | Identifiers | Additional information |
|---|---|---|---|---|
| Genetic reagent (C. elegans) | AVS752 | This paper | artEx41 | artEx41 [rab-3p::hpk-1::CFP+pCFJ90 (myo-2p::mCherry)]; hpk-1(pk1393); juIs76 [unc-25p::GFP+lin-15(+)] II |
| Genetic reagent (C. elegans) | AVS832 | This paper | rmIs132; artEx58 | rmIs132 [unc-54p::Q35::YFP] I; artEx58 [rab-3p::hpk-1+pCFJ90 (myo-2p::mCherry)] line 1 |
| Genetic reagent (C. elegans) | AVS833 | This paper | rmIs132; artEx59 | rmIs132 [unc-54p::Q35::YFP] I; artEx59 [rab-3p::hpk-1+pCFJ90 (myo-2p::mCherry)] line 2 |
| Genetic reagent (C. elegans) | AVS834 | This paper | rmIs132; artEx60 | rmIs132 [unc-54p::Q35::YFP] I; artEx60 [rab-3p::hpk-1+pCFJ90 (myo-2p::mCherry)] line 3 |
| Genetic reagent (C. elegans) | AVS835 | This paper | rmIs132; artEx61 | rmIs132 [unc-54p::Q35::YFP] I; artEx61 [rab-3p::hpk-1+pCFJ90 (myo-2p::mCherry)] line 4 |
| Genetic reagent (C. elegans) | AVS744 | This paper | rmIs132; artEx75 | artEx75 rab-3p::hpk-1(K176A)+pCFJ90 (myo-2p::mCherry); rmIs132 [Punc-54::Q35::YFP] I line 1 |
| Genetic reagent (C. elegans) | AVS748 | This paper | rmIs132; artEx79 | artEx79 rab-3p::hpk-1(D272N)+pCFJ90 (myo-2p::mCherry); rmIs132 [Punc-54::Q35::YFP] I line 1 |
| Genetic reagent (C. elegans) | AVS562 | This paper | rmIs132; artEx41 | rmIs132 [unc-54p::Q35::YFP] I; artEx41 [rab-3p::hpk-1::CFP+pCFJ90 (myo-2p::mCherry)] line 5 |
| Genetic reagent (C. elegans) | AVS563 | This paper | rmIs110; artEx41 | rmIs110 [F25B3.3p::Q40::YFP]; artEx41 [rab-3p::hpk-1::CFP+pCFJ90 (myo-2p::mCherry)] line 1 |
| Genetic reagent (C. elegans) | AVS837 | This paper | rmIs110; artEx92 | rmIs110 [F25B3.3p::Q40::YFP]; artEx92 [rab-3p::hpk-1::CFP+pCFJ90 (myo-2p::mCherry)] line 2 |
| Genetic reagent (C. elegans) | AVS838 | This paper | rmIs110; artEx93 | rmIs110 [F25B3.3p::Q40::YFP]; artEx93 [rab-3p::hpk-1::CFP+pCFJ90 (myo-2p::mCherry)] line 3 |
| Genetic reagent (C. elegans) | AVS839 | This paper | rmIs110; artEx94 | rmIs110 [F25B3.3p::Q40::YFP]; artEx94 [rab-3p::hpk-1::CFP+pCFJ90 (myo-2p::mCherry)] line 4 |
| Genetic reagent (C. elegans) | AVS214 | HC196 | | sid-1(qt9) V |
| Genetic reagent (C. elegans) | AVS265 | TU3401 | uIs69 | sid-1(pk3321)V; uIs69 [unc-119p::sid-1; myo-2p::mCherry]V |
| Genetic reagent (C. elegans) | AVS540 | This paper | rmIs132; uIs69 | rmIs132 [unc-54p::Q35::YFP] I; sid-1(pk3321)V; uIs69[unc-119p::sid-1; myo-2p::mCherry]V |
| Genetic reagent (C. elegans) | AVS541 | This paper | rmIs110; uIS69 | rmIs110 [F25B3.3p::Q40::YFP]; sid-1(pk3321)V; uIs69[unc-119p::sid-1; myo-2p::mCherry]V |
| Genetic reagent (C. elegans) | AVS713 | This paper | rmIs132; artEx41 | rmIs132 [unc-54p::Q35::YFP] I; artEx41rab-3p::hpk-1::CFP+pCFJ90 (myo-2p::mCherry)]; unc-31(e169)V |
| Genetic reagent (C. elegans) | AVS810 | This paper | rmIs132; artEx41 | rmIs132 [unc-54p::Q35::YFP] I; artEx41 [rab-3p::hpk-1::CFP+pCFJ90 (myo-2p::mCherry)]; unc-13(e51)I |
| Genetic reagent (C. elegans) | AVS84 | TJ375 | gpIs1 | gpIs1 [hsp-16.2p::GFP] IV |
| Genetic reagent (C. elegans) | AVS397 | This paper | gpIs1; artEx35 | gpIs1 [hsp-16.2p::GFP]; artEx35 [sur-5p::hpk-1::CFP+pCFJ90 (myo-2p::mCherry)] |
| Genetic reagent (C. elegans) | AVS399 | This paper | gpIs1; artEx33 | gpIs1 [hsp-16.2p::GFP]; artEx33 [rab-3p::hpk-1::CFP+pCFJ90 (myo-2p::mCherry)] |
| Genetic reagent (C. elegans) | AVS715 | This paper | artEx41; sqIs24 | artEx41 [rab-3p::hpk-1::CFP+pCFJ90 (myo-2p::mCherry)]; sqIs24 [rgef-1p::GFP::lgg-1+unc-122p::RFP] |
| Genetic reagent (C. elegans) | AVS716 | This paper | artEx41; sqIs13 | artEx41 [rab-3p::hpk-1::CFP+pCFJ90 (myo-2p::mCherry)]; sqIs13 [lgg-1p::GFP::lgg-1+odr-1p::RFP] |
| Genetic reagent (C. elegans) | AVS682 | This paper | rmIs132; artEx62 | artEx62 [tph-1p::hpk-1+pCFJ90 (myo-2p::mCherry)] line 1; rmIs132 [unc-54p::Q35::YFP] I |

*Continued*

| Reagent type (species) or resource | Designation | Source or reference | Identifiers | Additional information |
|---|---|---|---|---|
| Genetic reagent (*C. elegans*) | AVS683 | This paper | *rmIs132; artEx63* | *artEx63 [tph-1p::hpk-1+pCFJ90 (myo-2::mCherry)] line 2; rmIs132 [unc-54p::Q35::YFP] I* |
| Genetic reagent (*C. elegans*) | AVS685 | This paper | *rmIs132; artEx65* | *artEx65 [unc-47p::hpk-1+pCFJ90 (myo-2::mCherry)] line 1; rmIs132 [unc-54p::Q35::YFP] I* |
| Genetic reagent (*C. elegans*) | AVS686 | This paper | *rmIs132; artEx66* | *artEx66 [unc-47p::hpk-1+pCFJ90 (myo-2::mCherry)] line 2; rmIs132 [unc-54p::Q35::YFP] I* |
| Genetic reagent (*C. elegans*) | AVS691 | This paper | *rmIs132; artEx71* | *artEx71 [eat-4p::hpk-1+pCFJ90 (myo-2::mCherry)] line 1; rmIs132 [unc-54p::Q35::YFP] I* |
| Genetic reagent (*C. elegans*) | AVS692 | This paper | *rmIs132; artEx72* | *artEx72 [eat-4p::hpk-1+pCFJ90 (myo-2::mCherry)] line 2; rmIs132 [unc-54p::Q35::YFP] I* |
| Genetic reagent (*C. elegans*) | AVS693 | This paper | *rmIs132; artEx73* | *artEx73 [cat-2p::hpk-1+pCFJ90 (myo-2::mCherry)] line 1; rmIs132 [unc-54p::Q35::YFP] I* |
| Genetic reagent (*C. elegans*) | AVS694 | This paper | *rmIs132; artEx74* | *artEx74 [cat-2p::hpk-1+pCFJ90 (myo-2::mCherry)] line 2; rmIs132 [unc-54p::Q35::YFP] I* |
| Genetic reagent (*C. elegans*) | AVS809 | This paper | *sqIs13; artEx62* | *artEx62 [tph-1p::hpk-1+pCFJ90 (myo-2::mCherry)] line 1; sqIs13 [lgg-1p::GFP::lgg-1+odr-1p::RFP]* |
| Genetic reagent (*C. elegans*) | AVS794 | This paper | *sqIs13; artEx65* | *artEx65 [unc-47p::hpk-1+pCFJ90 (myo-2::mCherry)] line 1; sqIs13 [lgg-1p::GFP::lgg-1+odr-1p::RFP]* |
| Genetic reagent (*C. elegans*) | AVS709 | This paper | *artEx62* | *artEx62 [tph-1p::hpk-1+pCFJ90 (myo-2::mCherry)] line 1* |
| Genetic reagent (*C. elegans*) | AVS710 | This paper | *artEx65* | *artEx65 [unc-47p::hpk-1+pCFJ90 (myo-2::mCherry)] line 1* |
| Genetic reagent (*C. elegans*) | AVS888 | This paper | *aetEx99* | *artEx99 [unc-47p::hpk-1+pCFJ90 (myo-2::mCherry)] line 2* |
| Genetic reagent (*C. elegans*) | AVS816 | This paper | *sqIs13; artEx41* | *artEx41(rab-3p::hpk-1::CFP+pCFJ90 (myo-2:mCherry); sqIs13 [lgg-1p::GFP::lgg-1+unc-122p::RFP]; unc-30(ok613)* |
| Genetic reagent (*C. elegans*) | AVS872 | This paper | *sqIs13; artEx41* | *artEx41(rab-3p::hpk-1::CFP+pCFJ90 (myo-2:mCherry); sqIs13 [lgg-1p::GFP::lgg-1+unc-122p::RFP]; tph-1(mg280)* |
| Genetic reagent (*C. elegans*) | AVS836 | This paper | *artEx91* | *artEx91 [hpk-1p::GFP+tph-1p::mCherry] line 1* |
| Genetic reagent (*C. elegans*) | AVS757 | This paper | *artEx87* | *artEx87 [hpk-1p::mCherry+unc-47p::GFP] line 1* |
| Genetic reagent (*C. elegans*) | AVS001 | CB1370 | | *daf-2(e1370)* |
| Genetic reagent (*C. elegans*) | AVS022 | GR1309 | | *daf-2(e1370); daf-16(mgDf47)* |
| Genetic reagent (*C. elegans*) | AVS495 | CL6264 | *uIs60* | *uIs60 [unc-119p::YFP+unc119p::sid-1]; eri-1(mg366)* |
| Genetic reagent (*C. elegans*) | AVS817 | This paper | *uIs60; artIs1* | *artIs1 [sur-5p::HPK-1::CFP+pCFJ90 (myo-2p::m-cherry)]; uIs60 [unc-119p::YFP+unc119p::sid-1]; eri-1(mg366)* |
| Genetic reagent (*C. elegans*) | AVS488 | This paper | | *mxl-2(tm1516)* |
| Genetic reagent (*C. elegans*) | AVS1000 | This paper | *sqIs13; artEx65* | *artEx65 [unc-47p::hpk-1+pCFJ90 (myo-2p::mCherry)] line 1; sqIs13 [lgg-1p::GFP::lgg-1+odr-1p::RFP]; hlh-30(tm1978)* |
| Recombinant DNA reagent | pAVS1 | This paper | Plasmid | *rab-3p, hpk-1* cDNA, pPD95.75 backbone |
| Recombinant DNA reagent | pAVS2 | This paper | Plasmid | *dyp-7p, hpk-1* cDNA, pPD95.75 backbone |

*Continued on next page*

*Continued*

| Reagent type (species) or resource | Designation | Source or reference | Identifiers | Additional information |
|---|---|---|---|---|
| Recombinant DNA reagent | pAVS3 | This paper | Plasmid | *myo-3p, hpk-1* cDNA, pPD95.75 backbone |
| Recombinant DNA reagent | pAVS4 | This paper | Plasmid | *cat-2p, hpk-1* cDNA, pPD95.75 backbone |
| Recombinant DNA reagent | pAVS5 | This paper | Plasmid | *eat-4p, hpk-1* cDNA, pPD95.75 backbone |
| Recombinant DNA reagent | pAVS6 | This paper | Plasmid | *tph-1p, hpk-1* cDNA, pPD95.75 backbone |
| Recombinant DNA reagent | pAVS7 | This paper | Plasmid | *unc-47p, hpk-1* cDNA, pPD95.75 backbone |
| Recombinant DNA reagent | pAVS8 | This paper | Plasmid | *rab-3p, hpk-1(K176A)*, derived from pAVS1 |
| Recombinant DNA reagent | pAVS9 | This paper | Plasmid | *rab-3p, hpk-1(D272N)*, derived from pAVS1 |
| Recombinant DNA reagent | pPD95.75 | Addgene | Plasmid #1494 | |
| Sequence-based reagent | *atg-18F* | This paper | PCR primer | 5'-ACTTGAGAAAACGGAAGGTGTT |
| Sequence-based reagent | *atg-18R* | This paper | PCR primer | 5'-TGATAGCATCGAACCATCCA |
| Sequence-based reagent | *cdc-42F* | This paper | PCR primer | 5'-AGCCATTCTGGCCGCTCTCG |
| Sequence-based reagent | *cdc-42R* | This paper | PCR primer | 5'-GCAACCGCTTCTCGTTTGGC |
| Sequence-based reagent | *bec-1F* | This paper | PCR primer | 5'-TTTTGTTGAAAGAGCTCAAGGA |
| Sequence-based reagent | *bec-1R* | This paper | PCR primer | 5'-CAACCAGTGAATCAGCATGAA |
| Sequence-based reagent | *hpk-1F* | This paper | PCR primer | 5'-AGTATGCACAGCTCCATCAC |
| Sequence-based reagent | *hpk-1R* | This paper | PCR primer | 5'-CCATTATTGGGACCGGAACA |
| Sequence-based reagent | *xbp-1F* | This paper | PCR primer | 5'-TGCCTTTGAATCAGCAGTGG |
| Sequence-based reagent | *xbp-1R* | This paper | PCR primer | 5'-ACCGTCTGCTCCTTCCTCAATG |
| Sequence-based reagent | *gei-3F* | This paper | PCR primer | 5'-AAGTCCGAGTCGCTGAACAC |
| Sequence-based reagent | *gei-3R* | This paper | PCR primer | 5'- ATGCCTGAATGCTGACGCTC |
| Chemical compound, drug | Isopropylthiogalactoside (IPTG) | GoldBio | I2481c-100 | |
| Chemical compound, drug | TRIzol reagent | Life Technologies | Catalog: 15596026 | |
| Chemical compound, drug | FUdR | Fisher/Alfa Aesar | CAS 50-91-9 | |
| Chemical compound, drug | Aldicarb | Fluka Analytical | # 33386 | |

*Continued on next page*

*Continued*

| Reagent type (species) or resource | Designation | Source or reference | Identifiers | Additional information |
|---|---|---|---|---|
| Chemical compound, drug | Serotonin (5-HT) | Sigma | H9623-100mg | |
| Commercial assay kit | QuikChange II XL Site-Directed Mutagenesis Kit | Agilent | Catalog #200521 | |
| Commercial assay kit | RNeasy Plus Mini Kit | QIAGEN | Cat. No.:74034 | |
| Commercial assay kit | cDNA synthesis kit | Bio-Rad | #1708890 | |
| Commercial assay kit | PerfeCTa SYBR green FastMix | Quantabio | #101414-276 | |
| Commercial assay kit | TruSeq Stranded mRNA | Illumina | | |
| Software, algorithm | Prism | GraphPad | Version 7 | |
| Software, algorithm | AxioVision | | v4.8.2.0 | |
| Software, algorithm | FastQC | *Andrews, 2010* | | |
| Software, algorithm | Trimmomatic | *Bolger et al., 2014* | | |
| Software, algorithm | STAR 2.4.2a | *Dobin et al., 2013* | | |
| Software, algorithm | featureCounts | *Liao et al., 2014* | Version 1.4.6-p5 | |
| Software, algorithm | RSEM | *Li and Dewey, 2011* | | |
| Software, algorithm | R statistical software environment | *Team, 2013* | Version 4.0.2 | |
| Software, algorithm | DESeq2 | *Love et al., 2014* | | |
| Software, algorithm | GOSeq | *Young et al., 2010* | | |
| Software, algorithm | MuDataSeurat | *Hao et al., 2021* | | |

All strains generated in this study are available upon request from the Samuelson laboratory or can be found at the Caenorhabditis Genetics Center (https://cgc.umn.edu/).

All plasmids generated in this study are available upon request from the Samuelson laboratory.

All primers were generated at Integrated DNA Technologies (https://www.idtdna.com/pages).

## *C. elegans* strains and details

All strains were maintained at 20°C on standard NGM plates with OP50. For all experiments, animals were grown in 20× concentrated HT115 bacteria seeded on 6 cm RNAi plates. Details on the strains, mutant alleles, and transgenic animals used in this study are listed in the Key resources table. All strains generated in this study are available upon request from the Samuelson laboratory or can be found at the Caenorhabditis Genetics Center (https://cgc.umn.edu/).

## Generation of transgenic strains

To assemble tissue-specific constructs for increased *hpk-1* expression, the *hpk-1* cDNA was amplified and cloned under control of the following promoters: hypodermal *dpy-7p* (*Gilleard et al., 1997*), body wall muscle *myo-3p* (*Okkema et al., 1993*), pan-neuronal *rab-3p* (*Nonet et al., 1997*), dopaminergic neurons *cat-2p* (*Lints and Emmons, 1999*), glutamatergic neurons *eat-4p* (*Lee et al., 1999*), serotonergic neurons *tph-1p* (*Sze et al., 2000*), and GABAergic neurons *unc-47p* (*Gendrel et al., 2016*). In brief, promoter sequences were subcloned from existing plasmids into the *Prab-3p::hpk-1* plasmid. To create kinase domain point mutations *K176A* and *D272N*, the *Prab-3p::hpk-1* plasmid was used and mutations were performed with a QuikChange II XL Site-Directed Mutagenesis Kit (Agilent).

All the assembled plasmids were validated by sequencing prior to microinjection. These constructs were injected at 5 ng/ml together with *myo-2p::mCherry* at 5 ng/ml as co-injection marker and *pBlue-Script* at 90 ng/ml as DNA carrier.

## RNAi feeding

The *hpk-1* RNAi clone was originated from early copies of *Escherichia coli* glycerol stocks of the comprehensive RNAi libraries generated in the Ahringer and Vidal laboratories. The control empty vector (L4440) and *hpk-1* RNAi colonies were grown overnight in Luria broth with 50 μg/ml ampicillin and then seeded onto 6 cm RNAi agar plates containing 5 mM isopropylthiogalactoside (IPTG) to induce dsRNA expression overnight at room temperature (RT). RNAi clones used in this study were verified by DNA sequencing and subsequent BLAST analysis to confirm their identity.

## RT-qPCR analysis

For measurement of gene induction, animals were synchronized and grown at 20°C, then isolated at the specified age. For time course of *hpk-1* mRNA upregulation, animals were harvested at specified time points (D2, D4, D6, D8, and D10). For *hpk-1* mRNA regulation at D11 of adulthood, animals were synchronized, grown at 20°C, and harvested at D1 and D11 of adulthood. For autophagy genes upregulation, animals were synchronized, grown at 20°C, and harvested at D1 of adulthood. After harvesting animals, RNA extraction was followed by using TRIzol reagent (Life Technologies) followed by RNeasy Plus Mini Kit (QIAGEN). RNA concentration was measured using a Nanodrop, and RNA preparations were reverse transcribed into cDNA using the Bio-Rad cDNA synthesis kit (#1708890) as per the manufacturer's protocol. Quantitative real-time PCR was performed using PerfeCTa SYBR green FastMix (Quantabio) with three technical replicates for each condition. Primer sets with at least one primer spanning the exon were used to amplify the gene of interest. *cdc-42* mRNA levels were used for normalization. Fold-change in mRNA levels was determined using ΔΔ Ct method (*Livak and Schmittgen, 2001*). Primer sequences can be found in the Key resources table.

## Lifespan analysis

Traditional lifespan assays were performed essentially as described in *Das et al., 2017*; *Lee et al., 2003*. Briefly, animals were synchronized by egg prep bleaching followed by hatching in M9 solution at 20°C overnight. The L1 animals were seeded onto 6 cm plates with HT115 bacteria and allowed to develop at 20°C. At L4 stage, FUdR was added to a final concentration of 400 μM. Viability was scored every day or every other day as indicated in each figure. Prism 7 was used for statistical analyses and the p-values were calculated using log-rank (Mantel-Cox) method.

For lifespan with RNAi-mediated knockdown of TOR (*Figure 8—figure supplement 2*), animals for the *let-363* (TOR RNAi) condition were first raised on empty vector RNAi during development, and transferred to plates with *let-363* RNAi bacteria at L4 to avoid developmental arrest at L3. In addition to having IPTG present in the media of the RNAi plates, an additional 200 μl of 0.2 M IPTG was added directly to the bacterial lawn of all plates and allowed to dry immediately before adding animals. The assays were otherwise performed as just described.

For lifespan assays in *Figure 1F*, assay plates and animal populations were prepared as just described, and then lifespan was determined from analysis of time-series image data collected for a given animal at approximately hour intervals throughout adult life on modified Epson V800 flatbed scanners with our instance of the *C. elegans* 'lifespan machine' (*Oswal et al., 2021*; *Stroustrup et al., 2013*).

In all cases, scoring of viability was blinded with respect to genotype and values obtained at previous time points, and animals that died due to rupturing, desiccation on the side of the plate or well, or clear body morphology defects were censored.

## Thermotolerance assay

Survival assays at high temperature were conducted as previously described in *Johnson et al., 2014*. In brief, synchronized L1 animals were allowed to develop at 20°C and FUdR was added at the L4 stage. At day 1 adulthood, animals were moved to 35°C for a period of 6 and 8 hr. Animals were allowed to recover for 2 hr at 20°C, and viability was scored. In all cases, scoring of viability was blinded with respect to genotype and values obtained at previous time points; animals that died due to rupturing, desiccation on the side of the plate or well, or had clear body morphology defects were censored. Statistical testing between pairs of conditions for differences in the number of foci was performed using Student's t-test analysis.

## Induction of *hsp-16.2p*::*GFP* after heat shock

The *hsp-16.2p::GFP* animals were heat shocked on day 1 of adulthood for 1 hr at 35°C and imaged after 4 hr of recovery. Images were acquired using a Zeiss Axio Imager M2m microscope with AxioVision v4.8.2.0 software. The GFP a.u. values from the head and intestine of the animals were acquired using the area selection feature from AxioVision software. Two independent trials with a minimum of 20 animals per experiment were performed.

## Measurement of autophagosome formation using the *GFP::LGG-1/Atg8* reporter

*GFP::LGG-1/Atg8* foci formation was visualized as described in *Kumsta et al., 2017*. Briefly, L4 stage and day 1 adult animals were raised on HT115 bacteria at 20°C and imaged using a Zeiss Axio Imager M2m microscope with AxioVision v4.8.2.0 software at ×63 magnification. Two independent trials where at least 20 seam cells from 15 to 20 different animals were scored for *GFP::LGG-1* puncta accumulation. In all cases scoring of puncta was blinded with respect to genotype.

## Measurement of neurodegeneration in motor neurons

Animals carrying the *unc-25p::GFP* reporter strain to visualize VD and DD motor neurons were synchronized and allowed to develop at 20°C and FUdR was added at L4 stage. At L4, D2, and D9 of adulthood, animals were imaged in a Zeiss Axio Imager M2m microscope and the number of animals showing axonal breaks in the VD and DD motor neurons were scored. Axonal breaks were defined as an area of discontinued fluorescence in either the ventral or dorsal nerve cords.

## Polyglutamine aggregation in neurons and locomotion analyses

The visualization and quantification of the progressive decline in proteostasis in the nervous system (specifically the nerve ring) was performed as described in *Brignull et al., 2006*; *Lazaro-Pena et al., 2021*. In brief, synchronized *rgef-1p::polyQ40::YFP* L4 animals were treated with FUdR. Then, z-stack images from the nerve ring were acquired using a Zeiss Axio Imager M2m microscope with AxioVision v4.8.2.0 software at ×63 magnification. The fluorescent foci form compressed *z*-stack images from 20 animals per technical replicate were scored blind every other day from 4 to 8 days of adulthood.

To assess locomotion capacity, at day 2 of adulthood, 40 animals were transferred to a drop of 10 µl of M9 solution and the number of body bends performed in a period of 30 s from each animal was scored (as described in *Lazaro-Pena et al., 2021*). Statistical testing between pairs of conditions for differences in the number of foci was performed using Student's t-test analysis.

## Polyglutamine aggregation in muscle and paralysis analyses

The visualization and quantification of the progressive decline in proteostasis in muscle tissue was performed as described in *Lazaro-Pena et al., 2021*; *Morley et al., 2002*. Briefly, synchronized *unc-54p::Q35::YFP* L4 animals were treated with FUdR. Then, fluorescent foci from 20 animals per technical replicate were scored blind daily from days 1 to 3 of adulthood. To assess paralysis, at days 5, 7, 9, and 11 of adulthood, prodded animals that responded with head movement (and were therefore still alive) but no backward or repulsive motion were scored as paralyzed (as described in *Lazaro-Pena et al., 2021*). Statistical testing between pairs of conditions for differences in the number of foci was performed using Student's t-test analysis. Two to five independent transgenics lines were tested.

## Aldicarb assay

The aldicarb assay was performed as described in *Mahoney et al., 2006*; *Oh and Kim, 2017*. Plates containing aldicarb 1 mM were prepared the day before the assay and stored at 4°C. Animals were synchronized by picking at L4 stage and placing them in a plate with HT115 bacteria. After 24 hr, one aldicarb plate per strain were brought out of 4°C and a small drop of OP50 was placed in the center of the plate and let dry for 30 min. Approximately 25 animals were placed on each plate, and the number of paralyzed animals was scored every hour up to 5 hr. The assay was performed at RT.

## Exogenous 5-HT immobilization assay

Animals were synchronized by picking at L4 stage and placing them in a plate with HT115 bacteria. After 24 hr, serotonin (5-HT) was dissolved in M9 buffer to a 20 mM concentration (*Hart, 2006*).

Twenty worms were placed on a glass plate well with 400 µl of 20 mM serotonin for 10 min. The same number of control animals were placed in 400 µl of M9 buffer. The locomotion of worms (mobilized or immobilized) was annotated at the following time points: 2, 4, 6, 8, and 10 min. The assay was performed at RT.

## Sample preparation and sequencing for RNA-Seq

10 cm RNAi plates were seeded with 1 ml of 10× concentrated HT115 EV bacteria from an overnight culture, and allowed to dry for 1–2 days. Approximately 1000 synchronized L1 animals were added to each plate (~3000 animals per condition across multiple plates). At the L4 stage, 600 µl of 4 mg/ml FUdR was added. Animals were kept at 20°C for the duration of the experiment.

Day 2 adult animals were collected in M9 buffer, washed 2× in M9, and a final wash with DEPC-treated RNase-free water. Approximately two times the volume of the animal pellet of TRIzol reagent was added to each animal preparation, followed by brief mixing and freezing overnight at –80°C. Tubes were then allowed to partially thaw, and were vortexed for 5 min to assist with disrupting the cuticle. Samples were transferred to new tubes, and 200 µl of chloroform per 1 ml of TRIzol was added, followed by 20 s vortexing; the tubes settled at RT for 10 min. Supernatants were transferred to new tubes, and an equal amount of 100% ethanol was added and mixed before proceeding to column purification with the QIAGEN RNAeasy Mini kit according to the manufacturer's instructions. Samples were eluted with 30–50 µl of DEPC-treated water, and checked for initial concentration and quality with a Nanodrop ND-1000 spectrophotometer. Biological replicate samples were prepared from independently synchronized populations of animals.

Isolated RNA was provided to the University of Rochester Genomics Research Center for library preparation and sequencing. Prior to library preparation, RNA integrity of all samples was confirmed on an Agilent 2100 Bioanalyzer. Libraries for sequencing were prepared with the Illumina TruSeq Stranded mRNA kit, according to the manufacturer's instructions. Quality of the resulting libraries was checked again with a Bioanalyzer prior to sequencing to ensure integrity. Sequencing was performed on an Illumina HiSeq2500 v4, yielding an average of approximately 31 million single-ended 100 bp reads per sample. Quality of the output was summarized with FastQC (*Andrews, 2010*) and reads were trimmed and filtered with Trimmomatic to remove adapter sequence and any low-quality content occasionally observed toward the ends of reads (*Bolger et al., 2014*). After filtering out low-quality reads, an average of 30 million reads per sample remained, and were used for the rest of the analysis.

## Analysis of *hpk-1* null and wild-type control RNA-Seq dataset

RNA-Seq reads were aligned to the *C. elegans* genome (assembly WBcel235) with STAR 2.4.2a (*Dobin et al., 2013*) using gene annotation from Ensembl (version 82) (*Cunningham et al., 2022*). An average of 90.2% of reads per sample were uniquely and unambiguously mapped to the genome. These alignments were used as input to featureCounts (version 1.4.6-p5) for gene-level counts (*Liao et al., 2014*). Ambiguous or multi-mapping reads, comprising an average of approximately 9% of the reads per sample, were not included in the gene-level count results. Transcript-level quantification was also performed with RSEM (*Li and Dewey, 2011*) to obtain TPM (transcripts per million) expression estimates.

Further analysis was performed primarily in the R statistical software environment (version 4.0.2) (*Team, 2013*), utilizing custom scripts and incorporating packages from Bioconductor. Both count and TPM expression matrices were filtered to remove genes with low or no expression, keeping genes with at least 10 read counts in at least one sample. Genes were further filtered to keep only those annotated as protein coding, pseudogene, ncRNA, lincRNA, antisense, or snRNA, which could have been included in the sequenced libraries after poly-A selection; 96.3% of genes included in the final set are protein coding. Gene identifiers have been updated as necessary from newer versions of WormBase (WS284) for integration with recent resources and datasets.

Differential expression analysis was performed with DESeq2, and the optional fold-change shrinkage procedure was applied to moderate fold-changes for low-expression genes (*Love et al., 2014*). Genes significantly differentially expressed between *hpk-1(pk1393)* and N2 animals were taken as those with FDR-adjusted p-value less than 0.05 and absolute value of $\log_2$ fold-change of at least 1 (on a linear scale, down- or upregulated by at least twofold). Functional and over-representation analysis for differentially expressed genes was performed with GOSeq (*Young et al., 2010*) using

gene sets and pathways from the GO (*Harris et al., 2004*), KEGG (*Kanehisa and Goto, 2000*), and Reactome (*Jassal et al., 2020*).

## Integration with public tissue-specific and single-cell RNA-Seq datasets

### *C. elegans* Aging Atlas and CeNGEN single-cell datasets

We obtained differential expression results for old (days 8, 11, 15) vs young (days 1, 3, 5) animals across 200 cell clusters from the Aging Atlas single-cell RNA-Seq project (*Roux et al., 2022*). Significantly differentially expressed genes within each cluster were taken as those with adjusted p-value <0.05, and $\log_2$ fold-change magnitude of at least 0.5 (approximately 1.4 fold up- or downregulated on a linear scale). Additional neuron identity information was obtained by mapping Aging Atlas neuron clusters to clusters identified in late-larval stage animals in the CeNGEN single-cell neuronal transcriptome project (*Taylor et al., 2021*), assisted by correlations between clusters of the two datasets already provided with the Aging Atlas. Chemical signaling subtypes of neurons were determined by expression of marker genes commonly used in the field for fluorescent labeling of neurons: *tph-1* for serotonergic, *unc-47* for GABAergic, *eat-4* for glutamatergic, *cat-2* for dopaminergic, *cat-2* for cholinergic. Where a high correlation had been established between a cluster in the Aging Atlas and CeNGEN, a signaling type was assigned if the cluster expressed the associated marker gene at TPM greater than 75 based on the provided pre-normalized expression matrix available from the CeNGEN website. Ten Aging Atlas neuron clusters did not have an obvious corresponding cluster in the CeNGEN dataset; for these clusters neuron signaling class was based on marker genes having average cluster counts >0.5 across all cells in a cluster and, where possible, a consensus of the six Aging Atlas time points.

Normalized single-cell aging time course gene expression was obtained by processing raw counts in the H5AD file from the Aging Atlas resource website (http://c.elegans.aging.atlas.research.calico-labs.com/data). Briefly, the H5AD data from SCANPy was read in and converted into a Seurat object with MuDataSeurat (https://github.com/PMBio/MuDataSeurat) (*Hao et al., 2021*). SCTransform, as implemented in Seurat, was used to normalize raw counts for library size differences, yielding log-scale normalized expression values for each cell that were used for further analysis.

Genes uniquely expressed in neurons or significantly enriched compared to other tissues were obtained from tissue-specific bulk RNA-Seq reported by *Kaletsky et al., 2018*, due to the additional dynamic range provided over current single-cell sequencing technology.

### *C. elegans* kinome, TFs, neuronal genome, and longevity-associated genes

A collection of *C. elegans* kinases was assembled based on *Zaru et al., 2017*, and cross-referenced against current gene annotation in WormBase version WS284 (*Harris et al., 2020*) to keep only protein-coding genes and remove pseudogenes or 'dead' genes. This produced a list of 438 kinases in the *C. elegans* kinome (*Figure 2—figure supplement 4*, *Supplementary file 5*). *C. elegans* TFs were identified from a previously published compendium (*Nye et al., 1989*). Genes with known or predicted functions important to neurons, and their associated roles, were assembled from tables curated in *Hobert, 2013*. Genes with lifespan phenotypes primarily from perturbation studies in *C. elegans* utilizing RNAi or mutant strains were assembled from annotation in WormBase (WS282) as well as the GenAge database (*de Magalhães and Toussaint, 2004*).

## Statistical analysis for in vivo assays

Unless otherwise specified, GraphPad PRISM version 7 software was used for statistical analyses. Data were considered statistically significant when p-value was lower than 0.05. In figures, asterisks denote statistical significance as (*, $p<0.05$, **, $p<0.001$, ****, $p<0.0001$) as compared to appropriate controls. N number listed in figure legend indicates the number of one representative experiment among all biological trials.

## Acknowledgements

We would like to thank members of the Samuelson laboratory, past and present, for their thoughtful insight and assistance related to this project, especially Drs. Rachel Kitt and Sara Farrell. We would like to thank members of the Western New York Worm Group for their input and discussions. We

would like to thank Dr. Doug Portman (URMC) and his laboratory for strains and advice. Some strains were provided by the CGC, which is funded by NIH Office of Research Infrastructure Programs (P40 OD010440). We would like to personally thank Drs. Dirk Bohmann (URMC), Gary Ruvkun (Massachusetts General Hospital/Harvard Medical School), Arjumand Ghazi (University of Pittsburgh), Seung-Jae Lee (Korea Advanced Institute of Science and Technology), and Mary Wines-Samuelson for critical reading of this manuscript. We would like to thank Drs. Cynthia Kenyon and David Kelley (Calico Life Sciences) for providing us the full scRNA-Seq results for the old vs young differential expression analysis results for each cell cluster in the C. *elegans* Aging Atlas. We would like to thank Dr. John Ashton for assistance and the University of Rochester Genomics Research Center for conducting RNA-Seq. Research reported in this publication was supported by the National Institute on Aging of the National Institutes of Health under Award Number RF1AG062593. CADB was supported by F32HD105323. The content is solely the responsibility of the authors and does not necessarily represent the official views of the National Institutes of Health.

## Additional information

### Funding

| Funder | Grant reference number | Author |
|---|---|---|
| National Institute on Aging | RF1AG062593 | Maria I Lazaro-Pena<br>Andrew V Samuelson<br>Zachary C Ward<br>Adam B Cornwell |
| National Institutes of Health | F32HD105323 | Carlos A Diaz-Balzac |

The funders had no role in study design, data collection and interpretation, or the decision to submit the work for publication.

### Author contributions

Maria I Lazaro-Pena, Conceptualization, Resources, Formal analysis, Validation, Investigation, Visualization, Methodology, Writing - original draft, Writing – review and editing; Adam B Cornwell, Conceptualization, Resources, Data curation, Software, Formal analysis, Validation, Investigation, Visualization, Methodology, Writing - original draft, Writing – review and editing; Carlos A Diaz-Balzac, Resources, Validation, Investigation, Methodology, Writing – review and editing; Ritika Das, Conceptualization, Resources, Investigation, Visualization, Methodology, Writing – review and editing; Zachary C Ward, Investigation, Methodology; Nicholas Macoretta, Conceptualization, Resources, Formal analysis, Validation, Investigation, Methodology; Juilee Thakar, Conceptualization, Resources, Formal analysis, Supervision, Methodology, Project administration, Writing – review and editing; Andrew V Samuelson, Conceptualization, Resources, Formal analysis, Supervision, Funding acquisition, Validation, Investigation, Visualization, Methodology, Writing - original draft, Project administration, Writing – review and editing

### Author ORCIDs

Maria I Lazaro-Pena http://orcid.org/0000-0002-3061-8835
Adam B Cornwell http://orcid.org/0000-0002-0572-3107
Carlos A Diaz-Balzac http://orcid.org/0000-0002-4723-1282
Andrew V Samuelson http://orcid.org/0000-0002-3071-5766

### Decision letter and Author response

Decision letter https://doi.org/10.7554/eLife.85792.sa1
Author response https://doi.org/10.7554/eLife.85792.sa2

## Additional files

**Supplementary files**

• Supplementary file 1. Trials, data, strains, genotypes, number of animals, and statistics for lifespan data presented in *Figures 1a–d, f, 8b, f* , *Figure 8—figure supplement 2*, and *Figure 8—figure supplement 2*.

• Supplementary file 2. Experimental data, strains, genotypes, number of animals, and statistical analysis for data presented in *Figures 1e, 5c and e*.

• Supplementary file 3. Trials, data, analysis, strains, genotypes, number of animals, and statistics for thermotolerance data presented in *Figures 1g, 8a and e*.

• Supplementary file 4. Trials, data, analysis, strains, genotypes, RNAi conditions, and statistics for RT-qPCR expression analysis presented in *Figures 2a, e and 6l*.

• Supplementary file 5. Transcription factors differentially expressed in old vs young animals in the same cell clusters where *hpk-1* has significant expression changes with age in the Aging Atlas dataset. This includes a summary of which clusters exhibit upregulation and which exhibit downregulation for a transcription factor (TF), and if the gene has been previously shown to have aging or lifespan-associated phenotypes in RNAi or loss-of-function mutant experiments. Primary data was from the *C. elegans* Aging Atlas, and association to aging phenotypes was determined based on curated annotations from WormBase and GenAge. File is associated with *Figure 2*, *Figure 2—figure supplements 1 and 2*, *Figure 2—figure supplement 4*.

• Supplementary file 6. Kinases differentially expressed in old vs young animals out of any of the 200 clusters with DE analysis results from the Aging Atlas dataset. Out of 438 kinases identified in *C. elegans*, 186 have significant age-associated expression changes in at least one Aging Atlas cell cluster. For these kinases, we summarized the number of age-associated up- and downregulated changes, if they have been previously identified as having aging or lifespan phenotypes in loss-of-function experiments, and the fold-changes in *hpk-1* null vs WT animals for significant changes in our bulk RNA-Seq experiment. Primary data was from the *C. elegans* Aging Atlas, and association to aging phenotypes was determined based on curated annotations from WormBase and GenAge. File is associated with *Figure 2*, *Figure 2—figure supplement 1*, *Figure 2—figure supplement 2*, and *Figure 2—figure supplement 4*.

• Supplementary file 7. Experimental data, strains, genotypes, number of animals/cells, age, and statistical analysis for data presented in *Figure 3b–d*.

• Supplementary file 8. Experimental data, strains, genotypes, number of animals, hours on aldicarb, and statistical analysis for data presented in *Figure 3e*.

• Supplementary file 9. Experimental data, strains, genotypes, number of animals, minutes on 5-HT, and statistical analysis for data presented in *Figure 3f*.

• Supplementary file 10. Differential expression analysis results for RNA-Seq of *hpk-1(pk1393)* null mutant animals compared to wild-type, presented in this manuscript. *Sheet 1:* Differential expression for all 16,828 genes considered to be expressed in any sample in the experiment. *Sheet 2:* The subset of genes with significant expression changes between groups (adjusted p-value <0.05, log2 fold-change magnitude ≥1). Additional columns in both sheets include a summary of the raw and normalized expression for each group, and the human homologs from OrthoList, where applicable. File is associated with *Figure 4*.

• Supplementary file 11. Of the genes significantly differentially expressed in *hpk-1(pk1393)* null mutant animals vs wild-type, we found 280 overlap a set of genes with important functions in neurons (*Hobert, 2013*). For these genes, table includes a summary of their function, the differential expression fold-change and raw expression level in the *hpk-1(pk1393)* vs wild-type RNA-Seq comparison, and the clusters from the Aging Atlas dataset with significant age-associated expression changes (where applicable). File is associated with *Figure 4*.

• Supplementary file 12. Trials, data, analysis, strains, genotypes, number of animals, age, and statistical analysis of polyglutamine foci formation data presented in *Figure 5b, d, g, i, l*, *Figure 7a, c*, *Figure 5—figure supplement 1c,d*, *Figure 6a,b*, and *Figure 7—figure supplement 1a,c*.

• Supplementary file 13. Trials, data, analysis, strains, genotypes, number of animals, age, and statistical analysis of data for paralysis onset in animals expressing *polyQ::YFP* presented in *Figure 5h, j*, *Figure 7b, d*, *Figure 5—figure supplement 1e,f*, and *Figure 7—figure supplement 1b,d*.

• Supplementary file 14. Trials, fluorescent quantification, analysis, strains, genotypes, number of

animals, heat shock conditions, and statistical analysis of data presented in *Figure 6c and e*.

• Supplementary file 15. Trials, autophagosome quantification, analysis, strains, genotypes, number of cells, age, and statistical analysis of data presented in *Figure 6*, *Figure 8d, h, i, j*, and *Figure 8— figure supplement 1*.

• MDAR checklist

## Data availability

Our primary RNA-Seq data is available for review (GEO accession GSE220744: https://www.ncbi.nlm. nih.gov/geo/query/acc.cgi?acc=GSE220744). Data analysis scripts have been deposited at: https:// github.com/samuelsonlab-urmc/hpk1_manuscript_2023 (copy archived at *Cornwell, 2023*). All data generated or analyzed during this study are included in the manuscript and supporting files; Source data files have been provided for all figures and figure supplements in 14 supplementary files.

The following dataset was generated:

| Author(s) | Year | Dataset title | Dataset URL | Database and Identifier |
|---|---|---|---|---|
| Lazaro-Pena MI, Cornwell AB, Diaz-Balzac CA, Das R, Macoretta N, Thakar J, Samuelson AV | 2022 | Homeodomain-interacting protein kinase maintains neuronal homeostasis during normal *Caenorhabditis elegans* aging and systemically regulates longevity from serotonergic and GABAergic neurons | https://www.ncbi. nlm.nih.gov/geo/ query/acc.cgi?acc= GSE220744 | NCBI Gene Expression Omnibus, GSE220744 |

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

## Appendix 1

While single-cell RNA-Seq is a powerful approach already, well-documented challenges remain with respect to consistency in cell captures and differentiating biological noise from technical artifacts such as 'drop-outs' from low numbers of average reads per cell (*Hwang et al., 2018*; *Hwang et al., 2021*; *Ke et al., 2022*; *Nayak and Hasija, 2021*). We attempted to identify gene-gene correlations within cell clusters across individual ages based on the *C. elegans* Aging Atlas dataset and found that inconsistency in the number of cells per cluster across time points limits our ability to identify many robust and significant associations – such variability is likely technical in nature, as it is not globally consistent across cell clusters for a given time point. In most cases, when aggregated into 'young' and 'old' sample groups, as was done for differential expression analysis in the original Aging Atlas analysis (*Roux et al., 2022*) the groups represent comparable numbers of total cells for most clusters. The trends we have observed at the level of individual time points indicate that the age-associated upregulation of *hpk-1* does not always occur at the same time point across cell clusters and is not necessarily tightly linked to the upregulation of longevity-associated TFs in the same cluster (*Appendix 1—figure 1*). Thus, the age-associated upregulation of pro-longevity factors in many cell clusters may not be due to a highly coordinated regulatory process, but rather a distributed response to accumulating local stress or damage. More work will be necessary to resolve if such late-life expression changes indeed contribute to maintenance of wild-type lifespan and healthspan, and the mechanism of the changes thereof (e.g., increased transcription, downregulation of miRNA targeting, epigenetic changes) and the factors that mediate these changes.

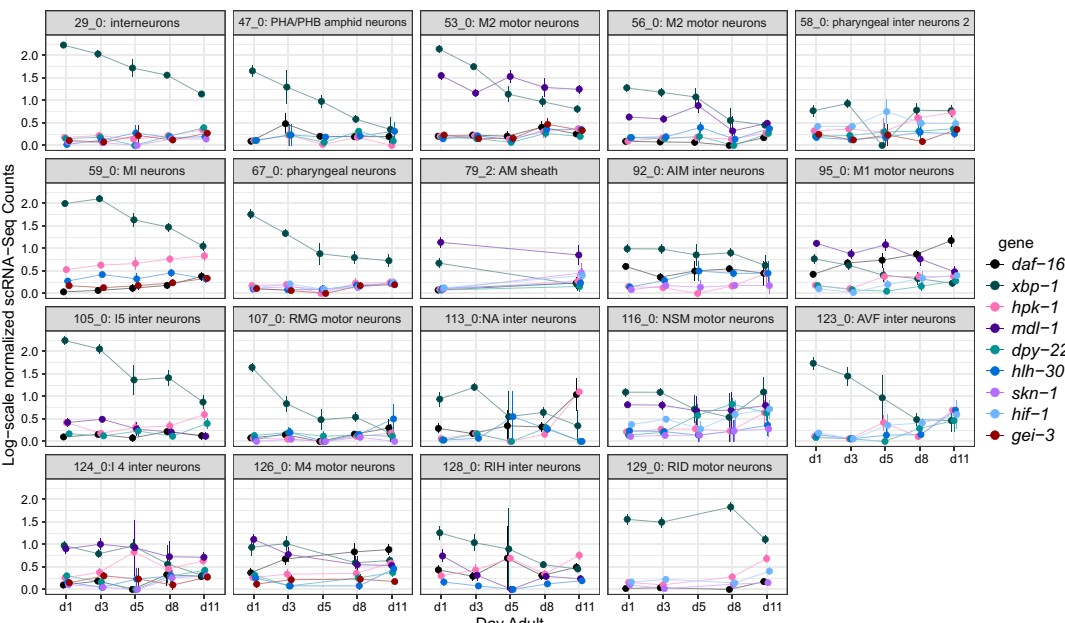

**Appendix 1—figure 1.** *hpk-1* and longevity-associated transcription factors are upregulated in overlapping cell clusters, but timing of the age-associated changes varies between cell clusters. Per-age expression summary for cell clusters with 4 or more differentially expressed longevity-associated transcription factors, of the 48 cell clusters in which hpk-1 is upregulated in old vs young animals. Each sub-plot shows log-scale normalized scRNA-Seq expression (mean and SEM) for the indicated cluster. Only genes which were significantly differentially expressed in the cluster are shown (old vs young animals, up- or downregulated). Primary data for this analysis was generated in the *C. elegans* Aging Atlas (*Roux et al., 2022*).

# Appendix 2

Loss of *hpk-1* upregulates expression of a number of neuronal genes including those involved in neuropeptide signaling, neurotransmission, release of synaptic vesicles, calcium homeostasis, and function of sensory neurons. In addition to upregulation of *ins* neuropeptides, 12 FMRFamide-related (flp) peptides, 19 neuropeptide-like proteins (*nlp*), 9 neuropeptide maturation genes (e.g., *egl-3* [proprotein convertase subtilisin/kexin type 2] and *egl-21* [carboxypeptidase E]), and 9 putative GPCR neuropeptide receptors were also significantly upregulated in *hpk-1* null mutants. Neurotransmitters are signaling molecules that trigger a response by binding to ionotropic or metabotropic neurotransmitter receptors in recipient neurons or tissues; we find that mainly the latter are upregulated in *hpk-1* null mutant animals, including acetylcholine (*gar-1, gar-2,* and *gar-3*), GABA (*gbb-2*), and glutamate (*mgl-3* and uncharacterized *C30A5.10*). In total, 9 metabotropic and 2 ionotropic glutamate receptors, 14 vesicular glutamate transport family genes, 9 acetylcholine receptor ligand-gated ion channels, and 9 GABA receptor ligand-gated ion channels were upregulated. Additionally, *ace-2, ace-3,* and *cha-1*, involved in acetylcholine degradation and synthesis (respectively), were upregulated. *comt-4*, a catechol-*O*-methyltransferase expected to be involved in the degradation of DA, was significantly downregulated.

Synaptic transmission, the formation and release of synaptic vesicles, carry neurotransmitters and rely on proteins necessary for docking, priming, and fusing to the axon terminal plasma membrane, which releases cargo into the synaptic cleft. Interestingly, loss of *hpk-1* causes the upregulation of synaptic proteins such as *unc-13*, *unc-10,* and *rab-3*, which are essential for vesicle priming, recruitment, and sequestration of vesicles, respectively. The process of vesicle release is highly dependent on a $Ca^{2+}$ signal detected by synaptotagmins proteins, which promotes vesicle fusion to the plasma membrane (*Deák, 2014*). In *hpk-1* null mutant animals, we observed upregulation of synaptotagmin *snt-1, snt-3,* and *snt-4* $Ca^{2+}$ sensor proteins for vesicle release.

Neuronal function is regulated by changes in calcium levels: a decrease in calcium uptake reduces neuronal function by a reducing neurotransmitter release (*Scriabine and Kazda, 1989*). In contrast, increasing calcium influx triggers activation of proteases and phospholipases, resulting in cell damage and cell death (*Cheung et al., 1986*; *Siesjö, 1989*). Voltage-gated $Ca^{2+}$ channels (VGCCs) are composed of 24 transmembrane alpha-1 subunits and associate with auxiliary beta-1 and alpha-2-delta subunits. *hpk-1* null mutant animals have increased expression of two of five alpha-1 subunits (*egl-19* and *unc-2*), and both alpha-2-delta subunits (*unc-36* and *tag-180*). Previous research describes the role of *egl-19* VGCC in the induction of the heat stress response (*Silva et al., 2013*): moderate upregulation of acetylcholine signaling improves muscle proteostasis by the activation of HSF-1, triggered by the *egl-19*-dependent $Ca^{2+}$ influx.

Neuronally expressed cilia are essential for cell motility, sensory perception, and signaling. The assembly and maintenance of sensory cilia depends on IFT: bidirectional movement of cargo to and from the base and tip of the axoneme. The *C. elegans* IFT machinery is composed of anterograde IFT motors, heterotrimeric kinesin-II and homodimeric OSM-3, retrograde dynein motors; and the IFT-A, -B, and BBSome particle complexes (*Hao et al., 2009*; *Scholey et al., 2004*). Loss of *hpk-1* induces the expression of a high number of IFT machinery components including *klp-11* (kinesin II motor), *che-3* (dynein motor), three and seven members of the IFT-A and -B complexes (respectively), and five members of BBSome complex. In mammals, there is a link between primary cilia signaling and aging-related brain disease, including neurodegenerative disorders (*Ma et al., 2022*).

