## [Editor Report]

This fundamental study substantially advances our understanding of how aging and stress resilience across an organism is determined by identifying a new player in this process and uncovering its mode of action. The evidence is solid as the methods, data and analyses broadly support the claims, with only minor weaknesses. The work will be of broad interest to the field of aging and protein homeostasis.

---

## [Decision Letter]

**Decision letter after peer review:**

Thank you for submitting your article "Homeodomain-interacting protein kinase maintains neuronal homeostasis during normal *Caenorhabditis elegans* aging and systemically regulates longevity from serotonergic and GABAergic neurons" for consideration by *eLife*. Your article has been reviewed by 2 peer reviewers, and the evaluation has been overseen by a Reviewing Editor and Carlos Isales as the Senior Editor. The reviewers have opted to remain anonymous.

Essential revisions:

1. You are encouraged to investigate whether core autophagy components (e.g., TFEB1) are required for the hpk-1 positive effects.

2. You need to evaluate whether hpk-1 levels increase with aging.

3. You should pay attention to their statistical analyses (see comments by Reviewer 2).

4. The main results from the scRNA-seq analysis should be independently validated (e.g. qPCR).

5. Body bends/sec over age should be measured to convincingly make the salient point.

6. Please use both the hpk-1 deletion strain and hpk-1 neuronal overexpression strain side-by-side in at least one of your assays.

*Reviewer #1 (Recommendations for the authors):*

I would first like to congratulate the authors on an impressive body of work. Below are a set of experiments and rationale:

hpk-1 can induce lgg-1 puncta in seam cells, but not intestine. The authors could test if hpk1 expression itself in either seam cell or intestine can autonomously induce lgg-1 puncta.

The use of lgg-1 puncta is informative, but what about using an autophagy reporter that measures actual flux?

Are the core autophagy components required for the hpk-1 positive effects? What about TFEB1?

Overexpression of hpk-1 in neurons is convincing ad show sufficiency, but what about tissue specific deletion of hpk-1 (FLP FRT) to assess opposite effects to measure necessity?

*Reviewer #2 (Recommendations for the authors):*

In the manuscript "Homeodomain-interacting protein kinase maintains neuronal homeostasis during normal *Caenorhabditis elegans* aging and systemically regulates longevity from serotonergic and GABAergic neurons", Lazaro-Pena et al., characterize neuronal phenotypes, longevity and stress -resilience in *Caenorhabditis elegans* to examine the contribution of homeodomain interacting protein kinase (HPK-1) to aging-associated phenotypes. They convincingly demonstrate that loss of hpk-1 increases neuronal dysfunction and detrimentally affects neuronal integrity, and thus hpk-1 is necessary to limit the motility-decline, stress resilience-decline and ventral and dorsal nerve cord-degeneration that normally occurs during aging; i.e. loss of hpk-1 exacerbates the decline, and rescuing hpk-1 in neurons prevents this exacerbation. They then conduct experiments to investigate how HPK-1 contributes to the maintenance of neurons during aging and show that hpk-1 overexpression in neurons (and in particular, serotonergic and GABAergic neurons) suppresses the accumulation of protein aggregates even in non-neuronal tissue, and activates protective stress responses such as autophagy and the chaperone induction. This leads them to conclude that HPK-1 activity maintains proteostasis, and thus preserves neuronal function and neuronal signaling required for the maintenance of tissue integrity during aging.

Overall, the data support a role for the role of HPK-1 in limiting the functional decline of neurons during aging. Several of the experiments support the authors conclusions. However, the manuscript has weaknesses that I believe need to be addressed, and further experimental evidence would be required to confirm the working model proposed by the authors.

1) Throughout the manuscript, the authors interchangeably utilize animals overexpressing hpk-1 in the nervous system [rab-3p::hpk-1 ], and hpk-1 deleted animals where hpk-1 is rescued in the nervous system [hpk-1(pk1393); rab-3p::hpk-1], to conduct experiments where they examine the contribution of hpk-1 to phenotype. However, one concern that arises with this approach is that it is unclear whether the activities of hpk-1 when overexpressed can be used to inform on the phenotypes that result when hpk-1 levels are insufficient. This is of particular importance, since the authors hypothesize (although they need to show this directly) that hpk-1 levels increase with aging, and that HPK-1 activity in neurons appears to affect neurons and non-neuronal tissue. In addition, there are several smaller details in the experimental design that do not support the strong conclusions that the authors make.

2) Several of the analyses are diffuse and are not supported by statistics.

3) The main results from the scRNA-seq analysis pertinent to the authors conclusions should be independently verified, as possible, using qPCR etc.

Specific details:

1) Figure 1: The authors state that HPK-1 can rescue age-associated decline in movement. This is read-out as body bends/sec in day 2 animals, and wild-type motility is compared to a hpk-1 deletion [hpk-1(pk1393)], and in a hpk-1 deleted animals where HPK-1 is re-expressed in the nervous system [hpk-1(pk1393); rab-3p::hpk-1]. There are two main problems I see with this experiment and the interpretation of the results. First, the authors are not showing 'age-associated' decline but are measuring body bends/sec at one time point, and that too in a young animals. Therefore, measurements of body bends/sec over age should be measured to convincingly make this point. Second, the authors show, convincingly that hpk-1 is necessary to maintain normal motility and resilience to heat stress, using the deletion mutant and by rescuing function by re-expressing hpk-1 in the nervous system. It would be interesting to see these experiments in the rab-3p::hpk-1 animals overexpressing hpk-1 in wild-type backgrounds. This would allow us to conclude whether hpk-1 is simply necessary, or whether its expression is sufficient to preserve function.

2) Figure 2: To investigate how hpk-1 mRNA expression changes with age, the authors rely on published scRNA-seq data. While the extensive analysis is useful, independent confirmation that hpk-1 mRNA levels increase with age using qRT-PCR is necessary. In addition, some mechanism to address whether HPK-1 activity also increases, or hpk-1 levels increase while activity remains the same or declines with aging would strengthen the manuscript.

3) Figure 2: The authors should evaluate whether the correlation of expression of hpk-1 mRNA with that of other transcription factors is significant. Correlation level, x-y plots, summary statistics of the correlation with significance are needed to make this point.

Moreover, the information in Figure 2, and later in Figure Supplement 4 are more confusing that supportive and could perhaps benefit from being presented in a more succinct and directed manner.

4) Figure Supplement 3: The experiments supporting the role of hpk-1 in the ILS pathway is confusing, and it is unclear what information can be gleaned from the epistasis studies at 20{degree sign}C versus 25{degree sign}C. In addition, in (G) the authors compare lifespans of a neuronal RNAi sensitive strain on empty vector and on daf-2 RNAi, with that of the neuronal RNAi sensitive strain systemically overexpressing hpk-1 and treated with daf-2 RNAi, to argue that neuronal HPK-1 and decreased ILS share overlapping mechanisms of longevity control. However, the lifespans of neuronal RNAi sensitive strain systemically overexpressing hpk-1 on empty vector alone needs to be included to support this conclusion. If these animals also showed that same lifespan extension as those on daf-2 RNAi, an alternative possibility is that HPK-1 and ILS extend lifespan through parallel pathways.

5) Figure 3 vs Figure 5 etc.: In addition to showing that hpk-1 is necessary to maintain neuronal integrity, using the deletion and rescued strains, the authors should evaluate the effects of the rab-3p::hpk-1 o/e strain in these experiments. In addition, the authors show that hpk-1 deletion mutants have decreased cholinergic neurotransmission as well as an increase in dopamine- and 5-HT-induced paralysis suggesting defects in neurotransmission as well as the ability to clear neurotransmitters from relevant synapses. Does overexpressing HPK-1 in neurons trigger the opposite phenotypes?

6) Line 327-33: There are no references cited for the dopamine- and 5-HT-induced paralysis treatments.

7) Figure supplement 5: The use of mutants in kinase activity is a strong positive control, but should be combined with some indication that these constructs (which are extrachromosomal, in my understanding) are expressed at the same levels as the rescuing hpk-1 overexpression construct. This is also true for the neuron specific rescue in Figure 8. The differences in the consequence of HPK-1 o/e in the different neurons would be strengthened if it was clear that the levels of expression are comparable.

8) Figure 5 and 6: the expression of HPK-1 in neurons is sufficient to decrease Q35 aggregation in muscle cells in the absence of stress, yet chaperone expression is only increased upon hsp-1 o/e after heat shock. The authors should elaborate on these results. Is HPK-1 activated under stress? Do the kinase deficient mutants not show this increase in hsp-16p::GFP?

9) Figure 6: lgg-1p::GFP::LGG-1 puncta are not an indication of autophagy unless coupled with flux measurements using brefeldin A.

10) Discussion and Figure 9: The authors should include tests for the effects of HPK-1 overexpression and deletion on 5-HT and GABA levels and activity during aging.

---

## [Author Response]

Essential revisions:1. You are encouraged to investigate whether core autophagy components (e.g., TFEB1) are required for the hpk-1 positive effects.

We thank Reviewer #1 for the suggestion and have done as requested. We tested whether loss of *hlh-30* (TFEB), *hsf-1* (HSF1), *pha-4* (FOXA), *mxl-2* (MLX), or *daf-16* (FoxO) are necessary for the induction of autophagy within hypodermal seam cells that occurs after increasing GABAergic expression of *hpk-1*, as of these transcription factors have previously been implicated in regulation of autophagy*.* Indeed, we find that *hlh-30, mxl-2, and daf-16* are all required for induction of autophagy when *hpk-1* is overexpressed solely in GABAergic neurons. In contrast, loss of either *hsf-1* or *pha-4* had no effect on autophagy induction; these new findings and the implications have been added to both the results and discussion in the revised manuscript and can specifically be found in *Figure 8J, Figure 8—figure supplement 1, Supplementary File 15*, and we have updated our model (*Figure 9B*), accordingly.

2. You need to evaluate whether hpk-1 levels increase with aging.

We thank Reviewer #2 for the suggestion and we independently found that *hpk-1* mRNA levels increase during normal aging of wild-type animals, which can be found in *Figures 2A,E,F* and *Supplementary File 4* in the revised manuscript. Please see Responses #17 and 20 for details

3. You should pay attention to their statistical analyses (see comments by Reviewer 2).

We thank Reviewer #2 for their suggestions. We address the reviewer’s concern in Responses #16 and #21. In regards to our analysis of the Aging Atlas dataset, it is also worth noting that we utilized more stringent statistical criteria for determining significant differential gene expression than was applied in the single-cell Aging Atlas preprint manuscript and associated web-based data portal (http://c.elegans.aging.atlas.research.calicolabs.com/embeddings). In particular, we used both a cutoff for adjusted p-value and a threshold on fold-change in order to determine the set of differentially expressed genes between young and old for each cell cluster.

4. The main results from the scRNA-seq analysis should be independently validated (e.g. qPCR).

We thank Reviewer #2 for the suggestion and have done as requested. We find that *hpk-1* expression increases during normal aging, confirmed this occurs within the nervous system in vivo, and via a candidate approach, also confirmed age-associated differential expression of two longevity-associated transcription factors. These results can be found in *Figure 2A,E,F* and *Supplementary File 4* in the revised manuscript. Please see Responses #17 and #20 for additional details.

5. Body bends/sec over age should be measured to convincingly make the salient point.

We thank the Reviewer #2 for the suggestion to test whether restoring HPK-1 within the nervous system is sufficient to rescue the accelerated age-associated decline in movement observed in *hpk-1* null mutant animals. We recorded age-associated changes in movement by imaging populations of wild-type, *hpk-1* null, and *hpk-1* null animals with neuronal *hpk-1* expression (*hpk1(0);rab-3p::hpk-1*) once every 15 minutes over the entire lifespan of the animals and converted these images into video files (spanning adult day 1 to 18). We find that restoring *hpk-1* within the nervous system is sufficient to rescue the age-associated accelerated decline in movement observed in the *hpk-1* null mutant animals. These results are provided in *Video 1* in the revised manuscript. Furthermore, neuronal *hpk-1* expression in the null background restores animals to wild-type lifespan. This further supports our conclusion that HPK-1 functions within the nervous system to maintain proteostasis, neuronal integrity and function during normal aging. We have included these findings within the revised manuscript (*Figure 1F*, and *Figure 2—figure supplement 1*).

6. Please use both the hpk-1 deletion strain and hpk-1 neuronal overexpression strain side-by-side in at least one of your assays.

We thank Reviewer #2 for this suggestion. Within the same experiment, we have previously demonstrated that loss of *hpk-1* impairs proteostasis and conversely *hpk-1* overexpression throughout the soma improves proteostasis, as measured both by age-associated accumulation of polyQ::YFP foci/aggregates within body wall muscle and the associated onset of paralysis occurring after collapse of the muscle proteome (*See Figure 2 of Das et al., 2017 PMID: 29036198*). In our current study we find analogous changes in proteostasis within both the nervous system and body wall muscle cells when *hpk-1* is either inactivated or overexpressed in the nervous system (*Figure 5* in this study). While these experiments were not conducted at exactly the same time, they were conducted in the same time frame and changes in proteostasis were compared to appropriate controls that are distinct (i.e., neuronal enhanced RNAi background for inactivation and wild-type background for overexpression).

Respectfully, we believe that we have sufficiently demonstrated that *hpk-1* null animals are shortlived, and *hpk-1* overexpression is sufficient to increase lifespan; repeating our lifespan analysis with multiple independent trials would require an unreasonable amount of time and effort for minimal gain to support the neuronal overexpression data in *Figure 1*. We and others have already demonstrated that *hpk-1* null mutant animals have a shortened lifespan, which is fully rescued by re-expression of *hpk-1* with the endogenous promoter but this rescue does not increase lifespan (*PMID: 26791749, PMID: 29036198*). Additionally, we previously found that loss of *hpk-1* within the nervous system is sufficient to shorten lifespan (*PMID: 29036198*). Thus HPK-1 activity within the nervous system is both necessary for a normal lifespan and overexpression is sufficient to extend longevity.

Collectively, between the current study and our findings published in *PMID: 29036198*, we have demonstrated:

– Lifespan: loss of *hpk-1* shortens, *hpk-1* overexpression increases. Further, within the nervous system, loss *hpk-1* shortens, whereas neuronal *hpk-1* overexpression increases.

– Protein homeostasis: loss of *hpk-1* impairs, *hpk-1* overexpression improves (polyQ::YFP in body wall muscle).

– Heat shock response: loss of *hpk-1* impairs, neuronal *hpk-1* overexpression enhances, heat shock stabilizes HPK-1.

– Autophagy: loss of *hpk-1* impairs induction, neuronal *hpk-1* overexpression induces autophagy.

– TORC1 signaling: TORC1 negatively regulates neuronal HPK-1, *hpk-1* is required for TORC1 inhibition to increase lifespan and induction of autophagy gene expression.

We agree with Reviewer #2 in principle; there is merit in side-by-side comparison of loss of function/null mutants and gain of function/overexpression in genetic analysis for any phenotype.

However, given the extensive analysis of the numerous aforementioned phenotypes by multiple members of the laboratory and limitations both in resources and personnel to repeat findings, we kindly disagree with the reviewer in the necessity to redo confirmatory time consuming assays.

Reviewer #1 (Recommendations for the authors):I would first like to congratulate the authors on an impressive body of work.

Thank you again!

Below are a set of experiments and rationale:hpk-1 can induce lgg-1 puncta in seam cells, but not intestine. The authors could test if hpk1 expression itself in either seam cell or intestine can autonomously induce lgg-1 puncta.

We thank the reviewer for the suggestion. We were surprised at a lack of intestinal phenotype as well. Only recently have we obtained intestinal proteostasis reporters, which we think will lend better insight into how HPK-1 activity (or lack thereof) impacts proteostasis and regulation of the proteostatic network within intestinal cells. We are in the process of performing the necessary crosses to assess cell autonomous and non-autonomous effects of *hpk-1* on intestinal cells.

Testing whether *hpk-1* overexpression in either the hypodermis or intestines is sufficient to induce autophagy is an interesting idea. However, we do not see *hpk-1* expression within intestinal cells post-development (nor after heat stress or TORC1 inactivation). While this is an area of interest we would like to explore more broadly, pursuit of these avenues is beyond the scope of the current study.

The use of lgg-1 puncta is informative, but what about using an autophagy reporter that measures actual flux?

We thank the reviewer for the suggestion. Rather than repeating assays with dual fluorescence autophagy reporters, LGG-1 mutant reporters that cannot be processed and conjugated to autophagosomes, or Bafilomycin A injection into *lgg-1p::GFP* animals, we hypothesized that as a transcriptional co-factor, HPK-1 regulates autophagy via controlling autophagy gene expression. Indeed, we found that neuronal HPK-1 overexpression is sufficient to induce *atg-18* and *bec-1* expression. We include these results in *Figure 6L* and *Supplementary File 4* in the revised manuscript. HPK-1 regulation of autophagy through changes in gene expression is consistent with our previous finding that *hpk-1* is required for the induction of autophagy gene expression after TORC1 (*daf-15*) inactivation (*PMID: 29036198*).

Are the core autophagy components required for the hpk-1 positive effects? What about TFEB1?

Please see Response #1 above.

Overexpression of hpk-1 in neurons is convincing ad show sufficiency, but what about tissue specific deletion of hpk-1 (FLP FRT) to assess opposite effects to measure necessity?

We thank the reviewer for this excellent suggestion; this is a high priority question we will address in follow up studies! As mentioned in comment #6, we previously demonstrated that loss of *hpk1* within the nervous system (using enhanced neuronal RNAi strains) shortens lifespan (*PMID: 29036198*). We are in the process of tagging endogenous *hpk-1* with a degron to allow us to utilize the auxin-inducible degron system and selectively remove HPK-1 with a high degree of spatial and temporal control, but we do not have this reagent yet. Characterizing and validating this new strain is beyond the scope of the current study, but will be an integral component of follow-up studies.

Reviewer #2 (Recommendations for the authors):In the manuscript "Homeodomain-interacting protein kinase maintains neuronal homeostasis during normal *Caenorhabditis elegans* aging and systemically regulates longevity from serotonergic and GABAergic neurons", Lazaro-Pena et al., characterize neuronal phenotypes, longevity and stress -resilience in *Caenorhabditis elegans* to examine the contribution of homeodomain interacting protein kinase (HPK-1) to aging-associated phenotypes. They convincingly demonstrate that loss of hpk-1 increases neuronal dysfunction and detrimentally affects neuronal integrity, and thus hpk-1 is necessary to limit the motility-decline, stress resilience-decline and ventral and dorsal nerve cord-degeneration that normally occurs during aging; i.e. loss of hpk-1 exacerbates the decline, and rescuing hpk-1 in neurons prevents this exacerbation. They then conduct experiments to investigate how HPK-1 contributes to the maintenance of neurons during aging and show that hpk-1 overexpression in neurons (and in particular, serotonergic and GABAergic neurons) suppresses the accumulation of protein aggregates even in non-neuronal tissue, and activates protective stress responses such as autophagy and the chaperone induction. This leads them to conclude that HPK-1 activity maintains proteostasis, and thus preserves neuronal function and neuronal signaling required for the maintenance of tissue integrity during aging.Overall, the data support a role for the role of HPK-1 in limiting the functional decline of neurons during aging. Several of the experiments support the authors conclusions. However, the manuscript has weaknesses that I believe need to be addressed, and further experimental evidence would be required to confirm the working model proposed by the authors.1) Throughout the manuscript, the authors interchangeably utilize animals overexpressing hpk-1 in the nervous system [rab-3p::hpk-1 ], and hpk-1 deleted animals where hpk-1 is rescued in the nervous system [hpk-1(pk1393); rab-3p::hpk-1], to conduct experiments where they examine the contribution of hpk-1 to phenotype. However, one concern that arises with this approach is that it is unclear whether the activities of hpk-1 when overexpressed can be used to inform on the phenotypes that result when hpk-1 levels are insufficient. This is of particular importance, since the authors hypothesize (although they need to show this directly) that hpk-1 levels increase with aging, and that HPK-1 activity in neurons appears to affect neurons and non-neuronal tissue. In addition, there are several smaller details in the experimental design that do not support the strong conclusions that the authors make.

We thank the reviewer for remarking on the limitations of genetic analysis of overexpression and loss of function. We appreciate that caution should be taken when using results from one to inform upon the other. We believe our overexpression analysis is informative as to the role of homeodomain interacting protein kinases in aging, and traditionally have been more cautious in drawing conclusions from null mutant phenotypes; as HPK-1 (and orthologs) have subtle roles in normal development. As mentioned in Response #13, this is one reason we are moving towards the use of a conditional *hpk-1* allele. Nevertheless, our conclusions are also supported from cellular readouts of HPK-1 activity, some of which we previously published and are supported by other studies (*PMID: 26791749, PMID: 29036198*). Between these and our current study we consistently observe converse phenotypes in *hpk-1* overexpression and loss of function mutant animals. We highlight findings in detail in Response #6. Our model is generally consistent with modes of regulation found from yeast to mammals, and that homeodomain interacting protein kinases are activated in response to a broad range of metabolic and stress signals; activation during normal aging is consistent with a role in mitigating age-associated stress. We were surprised to discover that HPK-1 was the most broadly upregulated kinase during normal aging, which is congruent with the analogous discovery of upregulation of longevity-associated transcription factors made by the Aging Atlas study. However, we agree the latter is a working model that will be tested more rigorously in follow-up studies, both by ourselves and others in the field.

2) Several of the analyses are diffuse and are not supported by statistics.

We thank the reviewer for helping us improve rigor. In the revised manuscript we have ensured that statistical analysis is included and described for all experiments.

3) The main results from the scRNA-seq analysis pertinent to the authors conclusions should be independently verified, as possible, using qPCR etc.

We thank the reviewer for the suggestion. We have independently verified results from our analysis of the Aging Atlas scRNA-seq dataset via RT-qPCR of RNA. Please see response #20 below for details.

Specific details:1) Figure 1: The authors state that HPK-1 can rescue age-associated decline in movement. This is read-out as body bends/sec in day 2 animals, and wild-type motility is compared to a hpk-1 deletion [hpk-1(pk1393)], and in a hpk-1 deleted animals where HPK-1 is re-expressed in the nervous system [hpk-1(pk1393); rab-3p::hpk-1]. There are two main problems I see with this experiment and the interpretation of the results. First, the authors are not showing 'age-associated' decline but are measuring body bends/sec at one time point, and that too in a young animals. Therefore, measurements of body bends/sec over age should be measured to convincingly make this point.

We thank the reviewer for the suggestion and have done as requested. Please see Response #5 above.

Second, the authors show, convincingly that hpk-1 is necessary to maintain normal motility and resilience to heat stress, using the deletion mutant and by rescuing function by re-expressing hpk-1 in the nervous system. It would be interesting to see these experiments in the rab-3p::hpk-1 animals overexpressing hpk-1 in wild-type backgrounds. This would allow us to conclude whether hpk-1 is simply necessary, or whether its expression is sufficient to preserve function.

We find that neuronal overexpression of *hpk-1* is sufficient to restore an apparent wild-type lifespan, but not extend it beyond wild-type (*Figure 1F* in the revised manuscript). This is in contrast to results with neuronal *hpk-1* overexpression in wild-type animals in which we observe an increase in lifespan (*Figure 1A*). We posit that HPK-1 activity in peripheral tissues is necessary to increase lifespan beyond wild-type, which is consistent with previous findings: inactivation of *hpk-1* with either intestinal, hypodermal, or neuronal cells is sufficient to shorten lifespan (*PMID: 29036198*). Whether tissue specific inactivation in peripheral tissues compromises cell intrinsic responses to age-associated stress, reception of neuronal signals, or subtly compromises development within those tissues to limit lifespan are unresolved questions, which we will address in follow-up studies using a conditional *hpk-1* allele. It is worth mentioning that whole animal *hpk-1* inactivation (feeding based RNAi) either solely during development or immediately post-development is sufficient to shorten lifespan (*PMID: 29036198*).

2) Figure 2: To investigate how hpk-1 mRNA expression changes with age, the authors rely on published scRNA-seq data. While the extensive analysis is useful, independent confirmation that hpk-1 mRNA levels increase with age using qRT-PCR is necessary. In addition, some mechanism to address whether HPK-1 activity also increases, or hpk-1 levels increase while activity remains the same or declines with aging would strengthen the manuscript.

We thank the reviewer for the excellent suggestion to increase reproducibility. Indeed, we independently verified that *hpk-1* mRNA increases during normal aging. First, we isolated agesynchronized animals at time points between days two and ten of adulthood and observed an age-associated increase in *hpk-1* expression. Next, we verified that age-associated changes in gene expression could be observed in two longevity-associated transcription factors identified in the Aging Atlas study. Consistent with findings from the Aging Atlas, we find significant reduction of *xbp-1* and increase in *gei-3* expression in older animals. Last, we confirmed that *hpk-1* is induced within the nervous system of older wild-type animals via a transcriptional *hpk-1* reporter (*hpk-1p::mCherry*). These results can be found in *Figure 2* and *Supplementary File 4* in the revised manuscript. We thank the reviewer for the additional suggestions, which will be a priority for follow up studies.

3) Figure 2: The authors should evaluate whether the correlation of expression of hpk-1 mRNA with that of other transcription factors is significant. Correlation level, x-y plots, summary statistics of the correlation with significance are needed to make this point.

We thank the reviewer for pointing out the nature of the expression association between *hpk-1* and the longevity-associated transcription factors was lacking in clarity. We investigated genegene correlations across time, based on re-analysis of the Aging Atlas dataset at a cell cluster level, but found that technical variability in the number of cells per cluster across timepoints limits identification of many significant associations. This variability is not globally consistent for a given time point across cell clusters, except at day 15 for which many fewer cells were captured in the original dataset. When cells are aggregated into “old” and “young” groups, which serves as the basis for the result in *Figure 2*, the groups represent comparable numbers of cells. As such, we recognize that these results represent genes which are similarly differentially expressed between the old and young groups, rather than strictly co-expressed, and we have revised our manuscript accordingly.

Interestingly, among the cell clusters in which *hpk-1* is upregulated with age and which have reasonable consistency in the cell counts across time points, we observe that the age at which upregulation of *hpk-1* and the longevity-associated TFs is first observed is not necessarily tightly linked. We posit that these changes are not being mediated by a single upstream regulatory process, but rather by multiple factors in response to accumulated physiological stress. We have taken a more conservative interpretation within the revised manuscript for greater precision and clarity.

Moreover, the information in Figure 2, and later in Figure Supplement 4 are more confusing that supportive and could perhaps benefit from being presented in a more succinct and directed manner.

We thank the reviewer for the suggestion and for the sake of clarity have moved *Figure 2D* to *Figure 2—figure supplement 3* in the revised manuscript. We prefer to retain *Figure 2—figure supplement 4* in the revised manuscript, we believe the graphic representation of significant age associated differential expression of all kinases creates a useful visual resource for future investigators. We have taken the reviewer’s advice and condensed the relevant text.

4) Figure Supplement 3: The experiments supporting the role of hpk-1 in the ILS pathway is confusing, and it is unclear what information can be gleaned from the epistasis studies at 20{degree sign}C versus 25{degree sign}C. In addition, in (G) the authors compare lifespans of a neuronal RNAi sensitive strain on empty vector and on daf-2 RNAi, with that of the neuronal RNAi sensitive strain systemically overexpressing hpk-1 and treated with daf-2 RNAi, to argue that neuronal HPK-1 and decreased ILS share overlapping mechanisms of longevity control. However, the lifespans of neuronal RNAi sensitive strain systemically overexpressing hpk-1 on empty vector alone needs to be included to support this conclusion. If these animals also showed that same lifespan extension as those on daf-2 RNAi, an alternative possibility is that HPK-1 and ILS extend lifespan through parallel pathways.

We thank the reviewer for highlighting this concern. We agree the inclusion of this data is not pertinent to the rest of the study, and have removed it from the revised manuscript.

5) Figure 3 vs Figure 5 etc.: In addition to showing that hpk-1 is necessary to maintain neuronal integrity, using the deletion and rescued strains, the authors should evaluate the effects of the rab-3p::hpk-1 o/e strain in these experiments. In addition, the authors show that hpk-1 deletion mutants have decreased cholinergic neurotransmission as well as an increase in dopamine- and 5-HT-induced paralysis suggesting defects in neurotransmission as well as the ability to clear neurotransmitters from relevant synapses. Does overexpressing HPK-1 in neurons trigger the opposite phenotypes?

We thank the reviewer for suggesting these experiments. While a straightforward idea, we believe rigorously evaluating whether overexpression of *hpk-1* alters or improves neurotransmission would require an extensive analysis of these phenotypes throughout *C. elegans* lifespan which would be time consuming and we note that neurotransmission is just one of many possible mechanisms. Identifying the mechanism(s) through which increased neuronal HPK-1 activity alters neuronal function to extend longevity is a high priority area for future studies, which we are pursuing through unbiased discovery-based approaches.

6) Line 327-33: There are no references cited for the dopamine- and 5-HT-induced paralysis treatments.

We thank the reviewer and have added the missing references.

7) Figure supplement 5: The use of mutants in kinase activity is a strong positive control, but should be combined with some indication that these constructs (which are extrachromosomal, in my understanding) are expressed at the same levels as the rescuing hpk-1 overexpression construct. This is also true for the neuron specific rescue in Figure 8. The differences in the consequence of HPK-1 o/e in the different neurons would be strengthened if it was clear that the levels of expression are comparable.

We thank the reviewer for the suggestion. Both kinase mutants (*hpk-1(K176A)* and *hpk1(D272N)*) were engineered via site-directed mutagenesis of wild-type *hpk-1* in the same plasmid backbone, which were used in *Figures 5b,d,g,I,l, 7a,c*. These HPK-1 constructs do not have fluorescent nor epitope tags, and we have not been able to observe *C. elegans* HPK-1 via immunoblotting using antibodies to mammalian orthologs (not shown).

All transgenic animals are created via injecting the same amount of both *hpk-1* plasmid (either wild-type or mutant) and co-injection marker. Pharyngeal mCherry expression of the co-injection marker (*myo-2p::*mCherry) is similar across all strains. In the revised manuscript we updated the methods section to describe the creation, sequence validation, and cataloguing of these reagents.

We considered measuring mRNA expression. However, the transgenic animals overexpressing either mutant or wild-type *hpk-1* have endogenous *hpk-1*, and it would be difficult to distinguish between endogenous and exogenous mRNA expression. While the *rab-3 promoter* drives strong neuronal expression, *hpk-1* is broadly expressed in the nervous system of adult animals, which compounds obtaining a clear result via FISH. A further complication is that while HPK-1 protein is not detected outside of the nervous system post-development, analysis of cell-type specific expression of *hpk-1* (Aging Atlas dataset) and our prior analysis of transgenic animals expressing *hpk-1* transcriptional reporters revealed the presence of *hpk-1* mRNA outside the nervous system in adult animals (*PMID: 29036198* and unpublished observations).

Determining whether *hpk-1* is expressed at similar levels when expressed in different neuronal sub-types (*Figure 7, Figure 7—figure supplement 1*) using different promoters is even more problematic; it is always possible threshold effects exist when using exogenous promoters. However, we do not believe the data supports this possibility for *hpk-1* overexpression in serotonergic and GABAergic neurons, as these neurons elicit subtype *specific* responses. However, negative results with *hpk-1* overexpression within either dopaminergic neurons (*cat2p*) or glutamatergic neurons (*eat-4p*) could theoretically be explained via a threshold effect; we have added a statement within the results to acknowledge this possibility. Nevertheless, these promoters are well-established for neuronal subtype overexpression.

Given the aforementioned complications and limitations in resources, time, and personnel, we have made revisions within the manuscript to acknowledge the caveat of our negative results in dopaminergic and glutamatergic neurons.

8) Figure 5 and 6: the expression of HPK-1 in neurons is sufficient to decrease Q35 aggregation in muscle cells in the absence of stress, yet chaperone expression is only increased upon hsp-1 o/e after heat shock. The authors should elaborate on these results. Is HPK-1 activated under stress? Do the kinase deficient mutants not show this increase in hsp-16p::GFP?

We thank the reviewer for asking for clarification. Specifically, we previously found that HPK-1 is induced both within the nervous system and hypodermal seam cells after heat shock through post-transcriptional mechanisms and that the heat shock response is compromised in *hpk-1* null mutant animals (*PMID: 29036198*). We have rephrased these findings within the discussion of the revised manuscript and removed text that obscured the point. We have not tested whether kinase deficient mutants alter the induction of the heat shock response. In follow-up studies we intend to conduct a comprehensive mutational and structure/function analysis to identify the regions and functions that are essential to mediate HPK-1 pro-longevity modalities.

9) Figure 6: lgg-1p::GFP::LGG-1 puncta are not an indication of autophagy unless coupled with flux measurements using brefeldin A.

We thank the reviewer for the suggestion. We assume the reviewer meant Bafilomycin A. Please see Response #11 above.

10) Discussion and Figure 9: The authors should include tests for the effects of HPK-1 overexpression and deletion on 5-HT and GABA levels and activity during aging.

We thank the reviewer for the suggestion but believe this is beyond the scope of the current study.